# MemoryField: Exploiting Gravitational Field for Long-term Memory Management

## Abstract

Despite the rapid progress of large language models (LLMs), which enables agents to perform complex decision-making and interaction, their limited long-term memory capacity hinders the effective retention and organization of historical interactions. This often leads to instability and semantic fragmentation in multi-turn dialogues and long-range reasoning tasks. Existing memory mechanisms struggle with structural reorganization, dynamic semantic retrieval, and the modeling of cognitive phenomena such as memory consolidation and forgetting. To address these challenges, we propose MemoryField, a novel dynamic spatial cognitive memory architecture driven by an attention-based gravitational field model. MemoryField represents memory items as nodes in a high-dimensional semantic space, where semantic attraction, repulsion, attention-driven forces, and decay mechanisms enable self-organized evolution and adaptive restructuring. By integrating node dynamics with fusion and forgetting processes, our approach ensures semantic coherence and cognitive stability. Extensive experiments demonstrate that MemoryField consistently outperforms existing memory mechanisms, improving dialogue quality by up to +4.9 Mauve and +3.3 ROUGE-L, boosting adversarial and temporal reasoning F1 by up to +14.7, and achieving superior performance across real-world tasks such as AlfWorld, ScienceWorld, HotPotQA, and FEVER, while maintaining strong cross-model generalization.

## 1 Introduction

The rapid advancement of artificial intelligence technologies has led to significant breakthroughs in large language models (LLMs) across natural language understanding, generation, and reasoning tasks (Vaswani et al., 2017; Chang et al., 2024). Consequently, LLM-based agents have emerged as a critical research focus in the field of AI (Guo et al., 2024; Xi et al., 2025). These agents possess autonomous decision-making and continuous interaction capabilities, enabling them to demonstrate substantial potential across a wide range of complex tasks (Cheng et al., 2024). In recent years, autonomous task agents such as AutoGPT (Yang et al., 2023) and BabyAGI (Nakajima, 2023), as well as reinforcement learning and knowledge-enhanced applications like Voyager (Wang et al., 2023a), Toolformer (Schick et al., 2023), and LangChain (Topsakal & Akinci, 2023), have showcased the powerful adaptability and task execution capabilities of LLM-driven agents in diverse environments.

Despite the strong performance of LLMs in short-term context modeling, their long-term memory capacity remains a critical limitation (Wang et al., 2023b). Specifically, LLM-based agents struggle to store and organize historical interaction data effectively and lack the ability to model long-term contextual continuity (Bulatov et al., 2022). This leads to instability, forgetting, and semantic discontinuities in multi-turn conversations, cross-task transfers, and long-term reasoning scenarios (Zhang et al., 2024). The absence of robust long-term memory mechanisms not only hinders the agent's ability to accumulate and reuse experience, but also limits its progression toward embodied intelligence or human-like cognitive capabilities (Wang et al., 2023a).

Currently, three primary approaches are being explored to address memory in LLMs: log-based memory, vector-based memory, and tool-augmented memory (Zhang et al., 2024). Log-based memory stores task histories or dialogue contents in chronological order, which is simple in structure but prone to redundancy and limited in revealing deep semantic relationships (Sordoni et al., 2015). Vector-based methods encode information into high-dimensional vectors and retrieve relevant content

based on similarity, enhancing relevance but lacking dynamic adjustment and semantic clustering capabilities—therefore struggling to support knowledge evolution and reasoning structure (Lewis et al., 2020). Tool-augmented memory relies on external knowledge bases or function calls to enhance functionality, but often neglects the optimization and self-evolution of internal memory structures (Nakano et al., 2021).

As a result, existing LLM memory mechanisms face three core challenges: the lack of structural reorganization capabilities, limited semantic retrieval efficiency, and an inability to effectively simulate key memory phenomena such as memory consolidation, conceptual fusion, and natural forgetting. To address these challenges, we propose a novel dynamic spatial cognitive memory architecture based on an attention-driven gravitational field model, and we are the first to manage memory in the form of a "force field." This framework constructs a quasi-physical interaction mechanism among memory nodes in high-dimensional semantic space, retaining the advantages of log-based memory accumulation while reconstructing the structure and access mechanisms of stored information. Each memory item is treated as a node in high-dimensional space, and four types of "forces" are designed—semantic attraction, repulsion, attentional center pull, and peripheral pushback. These forces guide the spatial reconfiguration and structural evolution of memory nodes based on semantic similarity, access frequency, and temporal decay.

Specifically, the attention gravitational field models memory state as a four-tuple $(C_i, P_i, V_i, A_i)$ representing content, position, velocity, and activation level, respectively. A complete set of physical evolution rules is defined to allow memory nodes to dynamically adjust their spatial layout during interaction. The system also incorporates node fusion (for conceptual abstraction and redundancy reduction) and a forgetting mechanism (for pruning long-term low-activity memory), alongside energy-based convergence control to ensure stability and manageability of the evolving memory topology.

Across dialogue, long-context reasoning, and real-world benchmarks, our framework demonstrates consistent advantages over both naive and advanced memory baselines. It improves multi-turn dialogue coherence, enhances reasoning stability under extended contexts, and achieves competitive performance in interactive environments such as AlfWorld, ScienceWorld, HotPotQA, and FEVER. Moreover, ablation analyses highlight the necessity of our proposed gravitational forces for ensuring semantic cohesion and interpretability. Taken together, these findings underscore the effectiveness, robustness, and generality of MemoryField as a scalable long-term memory solution for LLM-based agents.

- We propose MemoryField, an attention-driven gravitational memory architecture that models memory as particles in a high-dimensional semantic space. By integrating semantic attraction/repulsion, attention pull, fusion, and forgetting, it supports dynamic self-organization, abstraction, and natural forgetting for scalable long-term memory management.

- We validate MemoryField through extensive experiments on multi-turn dialogue, long-context reasoning, and real-world tasks, showing significant improvements in coherence, reasoning stability, and cross-model generality over strong baselines.

## 2 RELATED WORK

### 2.1 MEMORY MECHANISMS IN LLM-BASED AGENTS

With the widespread application of large language models (LLMs) in dialogue, reasoning, and task planning, agents have demonstrated the ability to solve complex tasks through long-term interactions (Vaswani et al., 2017; Wei et al., 2022; Wang et al., 2024; Xi et al., 2025). Efficient information management has thus become a core challenge, driving research into memory mechanisms for intelligent agents (Sumers et al., 2023; Guo et al., 2024). Early approaches mainly relied on limited context windows, which are insufficient for long-term and complex tasks (Brown et al., 2020; Touvron et al., 2023). Recent studies have proposed scalable long-term memory mechanisms, including skill storage, knowledge base construction, and dynamic updating strategies, as seen in systems like Voyager, AppAgent, and MemPrompt (Madaan et al., 2022; Wang et al., 2023a; Zhang et al., 2023). In addition, hierarchical memory models improve retrieval efficiency through summarization and aggregation (Lewis et al., 2020; Jiang et al., 2023). However, current methods are

still limited in dynamic adjustment and forgetting strategies, often relying on static mechanisms that struggle to balance information retention and redundancy elimination (Madaan et al., 2022; Liu et al., 2024; Cheng et al., 2024). Therefore, developing more flexible and dynamic memory management approaches has become an important trend.

## 2.2 DYNAMIC KNOWLEDGE ORGANIZATION AND FORCE FIELD MODELING

Inspired by particle interactions in physics, force-directed modeling has been widely used in graph optimization and the self-organization of complex networks (Fruchterman & Reingold, 1991; Eades, 1984; Kamada et al., 1989). The four-force equilibrium model utilizes attraction and repulsion mechanisms to enable adaptive adjustment among nodes, improving structural rationality and dynamics (Newman, 2003; Leskovec et al., 2007). In artificial intelligence, existing knowledge graphs (e.g., TransE) are mostly static and struggle to handle relational evolution and new knowledge generation (Bordes et al., 2013; Wang et al., 2017). Although dynamic knowledge graphs introduce temporal embeddings, their flexibility remains limited (Trivedi et al., 2017; Xu et al., 2020). Furthermore, current forgetting mechanisms are mostly static and cannot simulate cognitive phenomena such as associative reinforcement and natural forgetting (Ebbinghaus, 2013; Cai et al., 2018). These limitations highlight the urgent need for a knowledge organization method capable of dynamic adjustment, flexible restructuring, and cognitive forgetting.

For a comprehensive review, please refer to Appendix A.1.

## 3 METHOD

To improve memory organization in LLM-powered agents for long-term interaction and complex reasoning, we propose a dynamic spatial cognitive architecture driven by an attentional gravitational field. Memory nodes are modeled as particles in a high-dimensional Euclidean space $\mathbb{R}^n$, each containing a semantic content vector, position, velocity, and activity level. Through four types of forces—inter-node repulsion and attraction, and attraction and repulsion relative to the origin—combined with query-driven dynamics and time decay, the system supports nonlinear memory structures, self-organizing knowledge topologies, and cognitive phenomena such as reinforcement, abstraction, and forgetting. Figure 1 illustrates the framework of our constructed memory field.

### 3.1 MODEL ARCHITECTURE

In this system, each memory node is defined as $N_i = (C_i, P_i, V_i, A_i)$, where the meanings of each parameter are as follows:

$C_i \in \mathbb{R}^d$: Semantic content vector. It exists in the $d$-dimensional real number space and is used to represent the semantic information of the memory node. For example, in the text memory scenario, through word vectors or sentence vectors, the semantics of the text are transformed into numerical vector representations. Different semantic contents will correspond to different vector values, enabling the similarity between semantics to be measured through vector calculations, as shown in Figure 2.

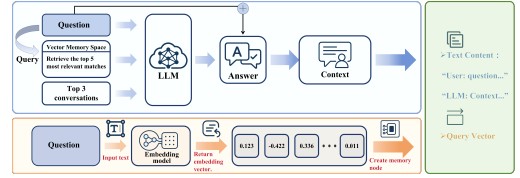

Figure 2: A user query is embedded and used to retrieve top-$k$ relevant memory nodes from the vector memory space. The retrieved nodes, along with the current question, are input into the LLM to generate an answer. The answer and its context are then stored as a new memory node, initializing semantic embedding, spatial position, and activity level for subsequent dynamic updates.

$P_i \in \mathbb{R}^n$: Spatial position. It is in the $n$-dimensional real number space and is used to determine the position of the memory node in the virtual space. This position information is crucial when simulating the interactions between nodes. For instance, the distance calculation between nodes depends on the position vectors, which in turn affect the attraction and repulsion forces between nodes.

$V_i \in \mathbb{R}^n$: Velocity. Also, in the $n$-dimensional real number space, it describes the movement speed of the memory node in space. The change in velocity is determined by the net force acting on the

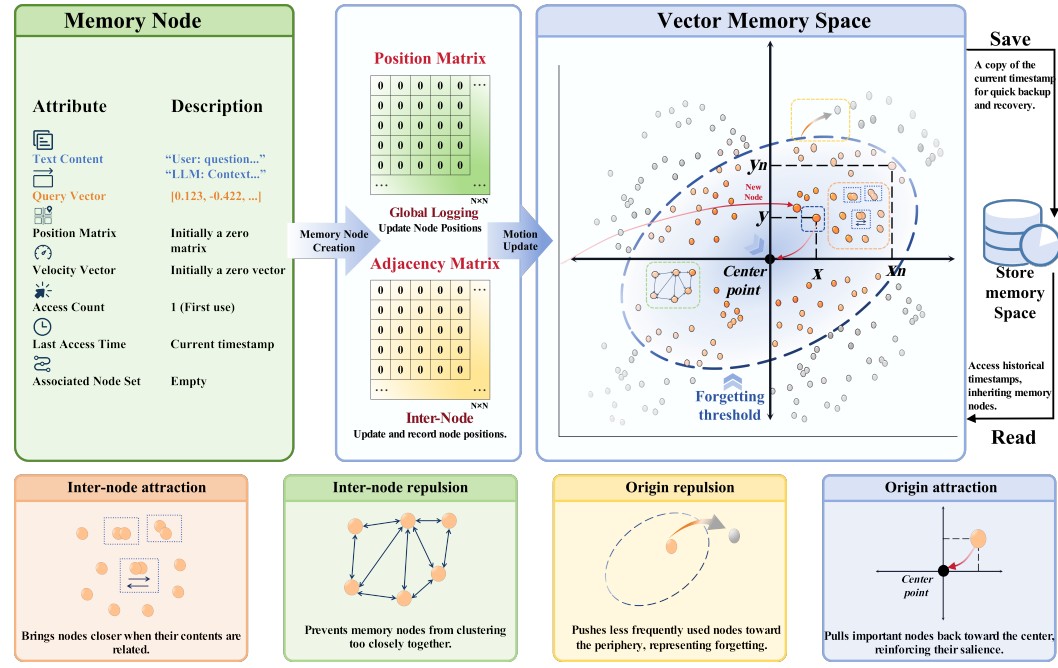

Figure 1: Overall workflow of the attentional memory system. Each memory node is modeled as a particle with semantic content, spatial position, velocity, and activity level. Node dynamics are governed by four forces: inter-node attraction and repulsion (based on semantic similarity and spatial proximity), and origin-based attraction and repulsion (driven by attention frequency and forgetting). The system maintains a self-organizing topology via position and adjacency matrices, supporting memory reinforcement, abstraction, and decay.

node and is closely related to the update of the position, reflecting the dynamic characteristics of the memory node in the system.

$A_i$: Activity level. It represents the degree of activity of a memory node and is used to determine whether the node will be forgotten. The activity level changes dynamically over time and with usage: for example, each time the node is accessed, its activity level increases; if it remains unaccessed for a prolonged period, the activity level gradually decays. When the activity level of a node falls below a certain threshold, the node is marked as forgotten, thereby releasing storage space and maintaining the efficiency of the memory structure.

Let $W_{ij} = f_{\text{sim}}(C_i, C_j)$ denote the semantic similarity matrix, where $f_{\text{sim}}$ is a function for calculating semantic similarity. It is calculated based on the semantic content vectors $C_i$ and $C_j$ of the nodes and reflects the degree of semantic association between two memory nodes. $D_{ij} = \|P_i - P_j\|$ is the spatial distance matrix, obtained by calculating the Euclidean norm of the position vectors of two nodes, and is used to measure the spatial distance between nodes. The net force on node $i$ is defined as:

$$F_i = F_{i,\text{repel}} + F_{i,\text{attract}} + F_{i,\text{origin - repel}} + F_{i,\text{origin - attract}} \tag{1}$$

This formula comprehensively considers four different types of forces, providing a comprehensive description of the force acting on the node in the system. The interaction of these forces determines the movement and state changes of the node. The following is an introduction to these four forces.

**Inter-node Repulsion:**

$$F_{i,\text{repel}} = \sum_{j \neq i} \alpha \cdot \frac{P_i - P_j}{\|P_i - P_j\|^3} \tag{2}$$

The inter-node repulsion is designed to prevent memory nodes from over-aggregating in space. When two nodes are close, the repulsion force increases, causing them to move away from each other. In the formula, $\alpha$ is the repulsion coefficient, which controls the strength of the repulsion force. $\frac{P_i - P_j}{\|P_i - P_j\|^3}$

indicates that the direction of the repulsion force is from node $j$ to node $i$, and the magnitude of the force is inversely proportional to the cube of the distance between the nodes. The closer the nodes are, the greater the repulsion force.

**Inter-node Attraction:**

$$F_{i,\text{attract}} = \sum_{j:W_{ij}>0} \beta_{ij} \cdot \frac{P_j - P_i}{\|P_j - P_i\|} \tag{3}$$

The inter-node attraction is used to connect semantically related nodes. Only when the semantic similarity matrix $W_{ij} > 0$, that is, when there is a certain semantic association between two nodes, will the attraction force be generated. $\beta_{ij}$ is the attraction coefficient related to nodes $i$ and $j$, which will be updated during operations such as associative queries. $\frac{P_j - P_i}{\|P_j - P_i\|}$ determines that the direction of the attraction force is from node $i$ to node $j$, and the magnitude of the attraction force is related to the attraction coefficient and the distance between the nodes.

**Repulsion from the Origin (Decay):**

$$F_{i,\text{origin - repel}} = \gamma_i \cdot \frac{P_i}{\|P_i\|^3} \tag{4}$$

**Parameter Updates.** This repulsion simulates the natural decay process of memory. $\gamma_i$ is the origin-repulsion coefficient related to node $i$, and $\frac{P_i}{\|P_i\|^3}$ indicates that the direction of the repulsion force is away from the origin, and the magnitude of the force is inversely proportional to the cube of the distance from the node to the origin. As the node moves away from the origin, the repulsion force gradually increases, meaning that the farther a node is from the origin, the more it is repelled, simulating the process by which memories that have not been accessed for a long time gradually weaken.

**Attraction to the Origin (Attention Frequency):**

$$F_{i,\text{origin - attract}} = \delta_i \cdot \|P_i\| \cdot \frac{-P_i}{\|P_i\|} \tag{5}$$

This attraction reflects the attention frequency of the node. $\delta_i$ is the origin - attraction coefficient related to node $i$, $\|P_i\|$ represents the distance from the node to the origin, and $\frac{-P_i}{\|P_i\|}$ determines that the direction of the attraction force is towards the origin. The closer a node is to the origin, the greater the attraction force it receives, indicating that nodes that are frequently accessed (with high activity levels) will be closer to the origin, reflecting the emphasis on frequently accessed memories.As shown in Figure 3.

Figure 3: Progressive construction of the position and adjacency matrices. As new memory nodes are added, semantic similarities and structural links are encoded to update global matrices, enabling spatial reasoning and interaction modeling.

During the operation of the system, some key parameters are updated based on different events.(1) $\delta_i(t + 1) = \delta_i(t) + \Delta\delta_{\text{direct}}$ (Direct Query): During a direct query operation, if a certain node is queried, its corresponding origin - attraction coefficient $\delta_i$ will increase by $\Delta\delta_{\text{direct}}$. This indicates that the node's degree of attention has increased due to the query, and the attraction force to the origin has strengthened. (2) $\beta_{ij}(t + 1) = \beta_{ij}(t) + \Delta\beta_{\text{assoc}}$ (Associative Query): When an associative query is performed and an association is found between nodes $i$ and $j$, the attraction coefficient $\beta_{ij}$ between them will increase by $\Delta\beta_{\text{assoc}}$, thereby strengthening the connection between semantically related nodes. (3) $\delta_i(t + 1) = \delta_i(t) \cdot (1 - \mu_\delta)$ (Time Decay): Over time, the origin-attraction coefficient $\delta_i$ will decay at a certain rate. $\mu_\delta$ is the decay rate. This simulates the phenomenon that even nodes that were frequently accessed in the past will gradually decrease in attention as time passes. (4) $\beta_{ij}(t + 1) = \beta_{ij}(t) \cdot (1 - \mu_\beta)$: Similar to $\delta_i$, the attraction coefficient $\beta_{ij}$ also decays over time, with $\mu_\beta$ being its decay rate, reflecting the weakening process of the semantic association between nodes.

**Association Probability (Establishment/Deletion).** $p_{\text{build}} = \sigma(w_1 \cdot \text{sim}(C_i, C_j) + w_2 \cdot (1 - D_{ij}/\theta_{\text{build}}))$: This is used to calculate the probability of establishing a new association. Here, $\sigma$ is

the Sigmoid function, which maps the input value to the interval $[0, 1]$, making the result conform to the range of probabilities. $w_1$ and $w_2$ are weight parameters used to adjust the relative importance of the semantic similarity $\text{sim}(C_i, C_j)$ and the spatial - distance - related term $(1 - D_{ij}/\theta_{\text{build}})$ in the probability calculation. $\theta_{\text{build}}$ is the distance threshold for association establishment. When the distance $D_{ij}$ between nodes is less than this threshold and the semantic similarity meets certain conditions, the probability of establishing an association will increase accordingly.

$p_{\text{drop}} = \sigma(v_1 \cdot (1 - \text{sim}(C_i, C_j)) + v_2 \cdot \frac{D_{ij} - \theta_{\text{build}}}{\theta_{\text{drop}} - \theta_{\text{build}}})$: This is used to calculate the probability of deleting an association. $v_1$ and $v_2$ are weight parameters, $(1 - \text{sim}(C_i, C_j))$ represents the semantic dissimilarity, $\frac{D_{ij} - \theta_{\text{build}}}{\theta_{\text{drop}} - \theta_{\text{build}}}$ is a distance - related term, and $\theta_{\text{drop}}$ is the distance threshold for association deletion. When the distance between nodes is greater than $\theta_{\text{drop}}$ or the semantic similarity is low, the probability of deleting the association will increase.

**Fusion.** (If $D_{ij} < \theta_{\text{fuse}}$ and $\text{sim}(C_i, C_j) > s_{\min}$) When multiple memory nodes meet the conditions that the distance is less than the fusion threshold $\theta_{\text{fuse}}$ and the semantic similarity is greater than the minimum similarity $s_{\min}$, a fusion operation will be performed:

$$C_f = f_{\text{fuse}}(C_1, \cdots, C_k), \quad P_f = \frac{\sum w_i P_i}{\sum w_i}, \quad V_f = \frac{\sum w_i V_i}{\sum w_i} \tag{6}$$

The semantic content vector $C_f$ after fusion is calculated by the function $f_{\text{fuse}}$, which comprehensively integrates the semantic information of each node participating in the fusion. The position vector $P_f$ and velocity vector $V_f$ are obtained by weighted averaging the corresponding vectors of the nodes participating in the fusion. The weights $w_i$ can be set according to actual situations, and usually, $w_i = 1$ is assumed for simple averaging. The fusion operation helps to reduce redundant memories and improve the organization and efficiency of memory.

**Activity Decay and Forgetting.** The activity decay formula is:

$$\text{Activity}(i, t) = \text{Activity}(i, t_0) \cdot \exp(-\lambda(t - t_0)) \tag{7}$$

This indicates that the activity level of the memory node decays exponentially over time. $\lambda$ is the decay coefficient, and $(t - t_0)$ is the time difference. As time increases, the activity level gradually decreases, reflecting the timeliness of memory. The forgetting judgment formula is:

$$\text{Forget}(i) = \begin{cases} \text{True}, & \text{if } \text{Activity}(i, t) < \theta_{\text{forget}} \text{ and } \|P_i\| > d_{\text{forget}} \\ \text{False}, & \text{otherwise} \end{cases} \tag{8}$$

When the activity level of a node is lower than the forgetting threshold $\theta_{\text{forget}}$ and the distance from the node to the origin is greater than the forgetting distance threshold $d_{\text{forget}}$, the node will be marked as a forgotten state. This mechanism ensures that memory nodes that have not been accessed for a long time and are far from the center of attention are properly processed, avoiding the occupation of excessive resources by invalid memories.

**Position Update.** The position and velocity of the node are updated based on the net force received: $V_i(t + \Delta t) = \beta \cdot V_i(t) + \alpha \cdot F_i(t) \cdot \Delta t$. This formula, based on the idea of Newton's second law, describes the update method of velocity. $\beta$ is the velocity decay coefficient, used to simulate the natural decay of velocity during movement; $\alpha$ is the coefficient related to the force, which controls the influence of the net force on the change in velocity; $F_i(t)$ is the net force on node $i$ at time $t$; $\Delta t$ is the time step. $P_i(t + \Delta t) = P_i(t) + V_i(t + \Delta t) \cdot \Delta t$: The position of the node is updated according to the updated velocity, reflecting the cumulative effect of velocity on position change.

## 4 EXPERIMENTS

### 4.1 EXPERIMENTAL SETUP

**Datasets.** We evaluate MemoryField on diverse benchmarks spanning dialogue, reasoning, and real-world tasks. For dialogue, we use Multi-session Chat (MSC), Conversation Chronicles (CC) and Very Long-Term Conversational (LoCoMo). For long-context reasoning, we construct five task categories—single-hop, multi-hop, temporal, open-domain, and adversarial—under context lengths from 4K to 16K. For real-world validation, we test on AlfWorld (sequential action execution),

ScienceWorld (scientific reasoning), HotPotQA (multi-hop QA), and FEVER (fact verification). Together, these benchmarks cover controlled settings and interactive environments.

**Models.** Our main experiments use GPT-3.5-turbo-16K, with and without MemoryField, and extend to GPT-4o, Claude Opus 4, LLaMA3.1-8B, Gemini 2.5 Flash, and Deepseek-R1. Baselines include All Dialogue History, All Memories + Context, Memory Retrieval, Rsum-LLM, MemoChat, COMEDY, and THEANINE, covering both naive and advanced memory mechanisms.

**Metrics.** For dialogue, we report BLEU-4, ROUGE-L, Mauve, and BERTScore. For reasoning, we measure F1 across the five task categories. For real-world tasks, we adopt official metrics: success rate (SR) for AlfWorld and HotPotQA/FEVER, and average reward (AR) for ScienceWorld. Cross-model evaluation follows automatic dialogue quality scoring.

**Implementation.** MemoryField is integrated as a structured *gravitational memory field*, where memory nodes evolve via attraction, repulsion, and decay forces. All methods use consistent prompts and fixed hyperparameters for fairness. Ablations disable individual forces to assess contributions. Repeated trials with fixed seeds ensure stable comparisons.

### 4.2 PERFORMANCE EVALUATION

Table 1: Performance of GPT-3.5-turbo-16K with and without MemoryField across different context lengths on MSC and CC tasks.

| Methods / Metrics | Multi-session Chat (MSC) | | | | Conversation Chronicles (CC) | | | |
|---|---|---|---|---|---|---|---|---|
| | BLEU-4 | ROUGE-L | Mauve | BERTScore | BLEU-4 | ROUGE-L | Mauve | BERTScore |
| All Dialogue History | 1.65 | 14.89 | 9.06 | 86.28 | 4.90 | 21.56 | 26.47 | 88.13 |
| All Memories & Current Context $\mathcal{D}$ | 1.56 | 14.89 | 10.62 | 86.23 | 4.41 | 20.00 | 31.86 | 88.02 |
| + Memory Update | 1.55 | 14.77 | 9.28 | 86.24 | 4.34 | 20.34 | 34.44 | 88.06 |
| Memory Retrieval | 1.92 | 15.49 | 11.16 | 86.20 | 4.40 | 20.48 | 33.24 | 88.09 |
| + Memory Update | 1.67 | 15.30 | 13.71 | 86.44 | 4.36 | 20.33 | 34.84 | 88.02 |
| Rsum-LLM | 0.75 | 11.53 | 2.45 | 84.61 | 0.98 | 11.42 | 2.28 | 85.59 |
| MemoChat | 1.42 | 13.11 | 7.72 | 85.94 | 2.31 | 15.87 | 15.12 | 87.03 |
| COMEDY | 1.06 | 12.79 | 7.27 | 85.29 | 1.70 | 13.57 | 19.55 | 85.90 |
| THEANINE | 1.80 | 15.37 | 18.62 | 86.70 | 6.58 | 22.68 | 64.41 | 88.58 |
| **MemoryField(Ours)** | 1.87 | **16.10** | **23.50** | **86.79** | **6.82** | **23.44** | **64.73** | **89.10** |

**Multi-turn Dialogue Evaluation.** As shown in Table 1, MemoryField achieves either the best or highly competitive overall performance across both dialogue datasets. On the MSC dataset, MemoryField reaches a Mauve score of 23.50, outperforming the best-performing baseline (THEANINE, 18.62) by 4.88 points. It also improves the ROUGE-L score to 16.10, representing a gain of approximately 3.3 points over COMEDY (12.79). In addition, MemoryField slightly surpasses other methods in both BLEU-4 and BERTScore. On the CC dataset, MemoryField yields modest improvements in BLEU-4 (6.82 vs. THEANINE's 6.58) and ROUGE-L (23.44 vs. 22.68), while maintaining a lead in Mauve (64.73 vs. 64.41). Notably, it achieves the highest BERTScore of 89.10, indicating superior semantic consistency. Compared to All Dialogue History and Memory Retrieval-based methods, MemoryField delivers an average improvement of more than 12 points in Mauve, and gains of 1.0–3.5 points in BLEU-4 and ROUGE-L. These results demonstrate the effectiveness of our structured gravitational memory field in enhancing semantic focus, reinforcing relevant information, and suppressing redundancy. Overall, MemoryField exhibits strong context preservation and improved generation quality, enabling more semantically coherent and consistent responses in multi-turn open-domain dialogue. These findings validate the model's memory advantages in long-range interactive scenarios.

**Very Long-term Dialogue Evaluation.** On the LoCoMo long-term dialogue QA benchmark, we fix the backbone model to gpt-4o-mini and compare four memory mechanisms: RSum-LLM, COMEDY, THEANINE, and MemoryField. To make the benefits of explicit memory clearer, we additionally include a Full Context baseline: for each question, we concatenate the entire dialogue history of the corresponding LoCoMo sample and feed it to the model without any explicit memory structure.

As shown in Table 2, the Full Context baseline, which is not constrained by the context window, achieves relatively strong F1 scores on the Single-Hop and Adversarial categories (around 41–68), but at the cost of processing roughly 17k tokens of context per question on average. In contrast, RSum-LLM compresses the dialogue via summarization and thus greatly reduces the effective context length (about 2k tokens), but suffers noticeable performance drops on Multi-Hop and Temporal questions, indicating that a purely linear summarization pipeline struggles to capture complex cross-session dependencies. Under a similar token budget, COMEDY performs slightly better overall than RSum-

| Method | Multi-Hop | | Temporal | | Open-Domain | | Single-Hop | | Adversarial | |
|---|---|---|---|---|---|---|---|---|---|---|
| | F1 | BLEU-1 | F1 | BLEU-1 | F1 | BLEU-1 | F1 | BLEU-1 | F1 | BLEU-1 |
| Full Context | 24.7 | 19.3 | 18.9 | 15.1 | 12.4 | 11.0 | 41.2 | 30.1 | 68.4 | 67.9 |
| RSum-LLM | 21.3 | 16.8 | 15.7 | 12.0 | 10.9 | 9.1 | 35.5 | 28.4 | 59.1 | 58.2 |
| COMEDY | 23.6 | 18.7 | 20.4 | 16.2 | 13.2 | 10.7 | 40.8 | 32.7 | 61.3 | 60.4 |
| THEANINE | 27.9 | 21.9 | 29.5 | 24.3 | 14.1 | 11.3 | 43.7 | 34.2 | 62.7 | 61.6 |
| MemoryField | 30.4 | 24.1 | 32.8 | 26.6 | 15.3 | 12.2 | 44.5 | 35.1 | 65.2 | 64.1 |

Table 2: Performance of different memory mechanisms on the LoCoMo long-term dialogue QA benchmark.

LLM, suggesting that compressive memory units provide a more robust representation for long-term conversations.

The timeline-based THEANINE performs particularly well on Multi-Hop and Temporal questions: in our results, its F1 scores are clearly higher than both the Full Context and the simple summarization baselines, while keeping the average context length around 2.2k tokens, which supports the intuition that timeline-structured memory is well-suited for long-range causal and temporal reasoning.MemoryField attains the best scores on the Multi-Hop, Temporal, and Adversarial categories. Overall, under a fixed backbone, equipping the agent with an appropriate long-term memory architecture not only substantially shortens the effective context window but also outperforms naïve full-context conditioning on challenging cross-session, multi-hop, and temporal reasoning questions.

**Long-context Reasoning Evaluation.** To further assess the effectiveness of MemoryField, we evaluate GPT-3.5-turbo-16K with and without MemoryField across various context lengths (4K to 16K) and five reasoning tasks. As summarized in Table 3, MemoryField consistently enhances model performance across all settings. Without memory augmentation, the model's F1 score improves with longer contexts (from 24.1 at 4K to 37.8 at 16K); however, it exhibits instability on complex tasks. Notably, in adversarial reasoning, the F1 score plummets from 13.1 to 2.1 at 16K, suggesting that extended contexts can introduce detrimental noise that impairs reasoning. In contrast, the MemoryField-enhanced model demonstrates improved stability and scalability. At 16K, it yields relative F1 improvements of 1.8 (single-hop), 2.7 (multi-hop), 14.7 (temporal), and 8.5 (adversarial), with an overall gain of 1.3. The gains are particularly substantial for temporal and adversarial tasks, highlighting MemoryField's effectiveness in handling long-range dependencies and semantic noise.

We attribute this improvement to MemoryField's mechanism of modeling past information as structured semantic entities, which are dynamically integrated via a gravitational attention mechanism. This mechanism amplifies relevant signals while suppressing irrelevant ones, enabling more robust and coherent reasoning paths across long contexts.

Table 3: F1 scores of GPT-3.5-turbo-16K with and without MemoryField across context lengths (4K–16K). Abbreviations: **S.H.** = Single Hop, **M.H.** = Multi Hop, **Temp.** = Temporal, **O.D.** = Open Domain, **Adv.** = Adversarial. Overall column is calculated as the average across all instances.

| Model | Ctx. | S.H. | M.H. | Temp. | O.D. | Adv. | Overall |
|---|---|---|---|---|---|---|---|
| GPT-3.5-turbo-16K | 4K | 31.7 | 25.4 | 16.8 | 27.6 | 13.1 | 24.1 |
| | 8K | 38.8 | 31.2 | 21.0 | 35.0 | 8.4 | 25.2 |
| | 12K | 51.1 | 40.4 | 25.0 | 36.5 | 6.4 | 33.5 |
| | 16K | 56.4 | 42.0 | 20.3 | 37.2 | 2.1 | 37.8 |
| **+MemoryField (Ours)** | **4K** | **33.4** | **27.8** | **23.3** | **34.4** | **24.7** | **26.8** |
| | **8K** | **40.3** | **34.5** | **28.3** | **39.7** | **19.6** | **28.6** |
| | **12K** | **54.2** | **42.9** | **41.5** | **40.2** | **17.2** | **35.6** |
| | **16K** | **58.2** | **44.7** | **35.0** | **41.9** | **10.6** | **39.1** |

**Cross-model evaluation.** To evaluate the performance of different memory mechanisms in multi-turn dialogue scenarios and to verify whether the proposed MemoryField can maintain consistent advantages across various mainstream large models (GPT-4o, Claude Opus 4, LLaMA3.1-8B, Gemini 2.5 Flash, Deepseek-R1), we conduct experiments on two standard benchmark datasets: MSC and CC. Both tasks involve long-range contexts and complex conversational dynamics. The baselines include history replay, memory-based retrieval mechanisms, summarization-based methods (Rsum-LLM), as well as existing memory-augmented models (MemoChat, COMEDY, THEANINE). The evaluation metric is ROUGE-L, where higher values indicate better dialogue generation quality.

As shown in Table 4, MemoryField achieves the best or highly competitive results across all models and both datasets. On the MSC dataset, MemoryField typically shows improvements of 0.2–1.0

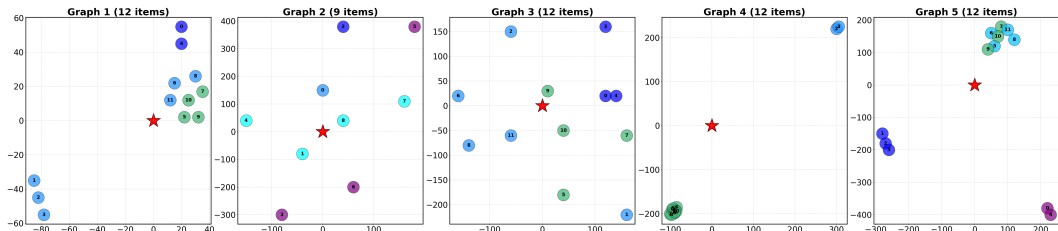

Figure 4: Visualization of memory node configurations: (a) baseline, (b) w/o node attraction, (c) w/o node repulsion, (d) w/o origin attraction, and (e) w/o full force mechanism.

over the best baseline. On the CC dataset, its advantage is even more pronounced, with average improvements of 1.0–2.5 compared to THEANINE and other methods. For instance, with GPT-4o, MemoryField reaches a score of 27.35 on CC, significantly surpassing THEANINE's 25.05; similar consistent gains are observed with Claude Opus 4 and LLaMA3.1-8B.

Table 4: Performance comparison across memory methods on MSC and CC tasks. Abbreviations of methods: **Hist.** = All Dialogue History, **Mem.+Ctx.** = All Memories & Context, **Retr.** = Memory Retrieval, **Rsum** = Rsum-LLM, **Memo** = MemoChat, **COM.** = COMEDY, **THEA.** = THEANINE, **MemField** = MemoryField (Ours).

| Model Name | Task | Hist. | Mem.+Ctx. | Retr. | Rsum | Memo | COM. | THEA. | MemField |
|---|---|---|---|---|---|---|---|---|---|
| GPT-4o | MSC | 18.25 | 18.32 | 17.80 | 14.30 | 15.10 | 14.15 | 16.90 | **18.72** |
| GPT-4o | CC | 24.15 | 23.75 | 23.40 | 14.90 | 18.10 | 16.25 | 25.05 | **27.35** |
| Claude Opus 4 | MSC | 17.90 | 18.12 | 17.30 | 13.80 | 14.65 | 14.90 | 16.10 | **17.90** |
| Claude Opus 4 | CC | 23.85 | 22.45 | 23.10 | 14.45 | 17.25 | 15.95 | 24.90 | **26.10** |
| LLaMA3.1 8B | MSC | 16.90 | 17.50 | 18.00 | 13.10 | 14.20 | 14.45 | 15.70 | **17.50** |
| LLaMA3.1 8B | CC | 23.05 | 21.75 | 22.15 | 13.40 | 16.20 | 15.00 | 23.50 | **25.05** |
| Gemini 2.5 flash | MSC | 17.15 | 17.80 | 17.20 | 13.50 | 14.35 | 14.05 | 15.60 | **17.40** |
| Gemini 2.5 flash | CC | 23.35 | 22.25 | 22.80 | 14.55 | 17.00 | 16.00 | 24.10 | **25.85** |
| Deepseek-R1 | MSC | 17.25 | 17.65 | 17.10 | 13.80 | 14.50 | 14.30 | 15.90 | **17.60** |
| Deepseek-R1 | CC | 23.25 | 22.05 | 22.45 | 14.80 | 16.75 | 15.60 | 24.25 | **25.60** |

Traditional summarization-based methods (e.g., Rsum-LLM) and some earlier memory models (e.g., COMEDY, MemoChat) perform relatively poorly in long dialogue settings, struggling to capture global context. While THEANINE demonstrates competitive performance in certain cases, it still falls short of MemoryField. Importantly, MemoryField delivers stable improvements across diverse model architectures, indicating that its memory mechanism possesses strong generality and transferability.

### 4.3 ABLATION STUDY

As illustrated in Figure 4, we visualize the spatial distribution of memory nodes under different force configurations. When all four forces—node attraction, node repulsion, origin attraction, and origin repulsion—are enabled (Figure 4a), the nodes form a well-structured and coherent layout around the central query point (red star). Node attraction clusters semantically related items, node repulsion prevents overlap, origin attraction pulls important nodes toward the center, and origin repulsion ensures dispersion. Their synergy yields semantically cohesive and spatially interpretable memory organization.

In contrast, disabling all forces (Figure 4b) produces a random distribution, where nodes scatter without clear semantic clustering and some drift far from the query. This highlights the necessity of the gravitational field mechanism for generating meaningful and interpretable memory structures.

To further analyze the role of each force, we conduct ablation experiments (Figures 4c–e). Removing node attraction disrupts semantic clustering, yielding more uniform but less coherent layouts, showing its importance for encoding semantic similarity. Disabling node repulsion collapses nodes into dense clusters, confirming its role in maintaining separation and preventing crowding. Without origin attraction, local clusters still form, but the global structure drifts away from the query point, indicating its importance for contextual alignment. Collectively, these results demonstrate that each force contributes uniquely to the memory topology, and their combination is essential for achieving a balanced, interpretable, and effective memory organization.

### 4.4 REAL-WORLD TASK EVALUATION

To assess practical effectiveness, we evaluate MemoryField on several real-world benchmarks, targeting environments that require long-horizon reasoning, interactive decision-making, and evidence

verification, and ask whether it delivers consistent performance gains. We consider four representative real-world tasks: (1) AlfWorld, a household environment requiring sequential action execution, evaluated with success rate (SR); (2) ScienceWorld, a scientific experiment environment requiring reasoning and multi-step tool usage, evaluated with average reward (AR); (3) HotPotQA, a multi-hop question answering benchmark measuring reasoning accuracy (SR); (4) FEVER, a fact verification task evaluating evidence retrieval and logical consistency (SR). We compare MemoryField against multiple baselines, including zero-shot reasoning (Z-CoT, F-CoT, CoT-SC), interactive decision-making approaches (SayCan, ReAct), and reflection-based reasoning (Reflexion).

**Objective.** It is important to note that the experimental setup is not intended to directly compare MemoryField against competing methods. Instead, under a fixed agent backbone and planning framework, we replace only the memory mechanism to evaluate the effectiveness of MemoryField on real-world datasets and agent-planning tasks. ReAct and other reasoning methods can be combined with different memory modules (including MemoryField), and the two are complementary in design rather than mutually exclusive.

**Evaluation Setting.** We define the Success Rate as the proportion of episodes in which the agent produces the correct final answer or completes the task objective. For QA tasks (HotPotQA, FEVER), a prediction is correct if it matches the gold answer under the standard Exact Match normalization (lowercasing, punctuation removal, and whitespace normalization). For HotPotQA, the agent retrieves up to $K = 5$ candidate paragraphs per step and is allowed $T = 3$ reasoning-retrieval iterations. The final answer is generated by the LLM and evaluated using EM. For FEVER, the agent interacts with the evidence retrieval environment and outputs one of three labels (Supported / Refuted / NotEnoughInfo). SR is computed as the proportion of correct labels, with a maximum of $T = 3$ retrieval calls. For embodied environments like AlfWorld, SR indicates successful task completion, while for ScienceWorld, the Average Reward measures multi-step reasoning and tool use, reflecting the agent's performance over all episodes. Details are shown in Appendix A.6.

Table 5: Performance comparison of reasoning and memory-augmented methods across multiple real-world benchmarks. Metrics: **SR** = Success Rate, **AR** = Average Reward. "-" means not reported.

| Method | AlfWorld (SR%) | ScienceWorld (AR) | HotPotQA (SR%) | FEVER (SR%) |
|---|---|---|---|---|
| Z-CoT | - | - | 0.01 | 0.39 |
| F-CoT | 0.43 | 16.58 | 0.32 | 0.61 |
| CoT-SC | 0.57 | 15.24 | 0.33 | 0.62 |
| SayCan | 0.60 | 12.36 | - | - |
| ReAct | 0.57 | 15.05 | 0.34 | 0.63 |
| Reflexion | 0.71 | 19.39 | 0.39 | 0.68 |
| **MemoryField (Ours)** | **0.75** | **20.42** | **0.41** | **0.71** |

**Results and Analysis.** As in Table 5, MemoryField achieves the best or highly competitive results across all benchmarks. In AlfWorld, MemoryField attains a success rate of 0.75, outperforming Reflexion (0.71) and demonstrating stronger robustness in long-horizon action planning. In ScienceWorld, MemoryField achieves an average reward of 20.42, exceeding Reflexion (19.39), which highlights its advantage in scientific reasoning and tool usage. In HotPotQA, MemoryField obtains 0.41, surpassing all baselines and showing its ability to maintain consistency in multi-hop reasoning. In FEVER, MemoryField reaches 0.71, higher than Reflexion (0.68), confirming its benefit in fact verification tasks. These results verify that MemoryField consistently improves agent performance in diverse real-world scenarios, demonstrating its strong generalization ability and robustness under challenging interactive and reasoning tasks.

## 5 CONCLUSION

In this paper, we propose MemoryField, a novel attention-driven gravitational memory architecture designed to address the challenges of long-term memory management in LLM-based agents. By modeling memory nodes as particles in a high-dimensional semantic space and simulating their dynamic evolution through force-directed interactions (semantic attraction, repulsion, attention-centric pull, and decay), we achieve structured memory self-organization, conceptual abstraction, and natural forgetting. Extensive experiments on multi-turn dialogue and long-context reasoning benchmarks demonstrate that, compared with traditional vector-based and graph-augmented memory methods, MemoryField significantly improves semantic coherence, information retention, and reasoning consistency. These results validate the potential of MemoryField in long-term interaction and adaptive knowledge management.

REPRODUCIBILITY STATEMENT

We are committed to the full reproducibility of this work. The proposed MemoryField architecture, including the gravitational force–driven memory dynamics and update rules, is described in detail with pseudocode in the Appendix, ensuring that future researchers can directly reproduce and extend our study. Our experimental setup is comprehensively introduced in Section 4. Details of hyperparameter choices, ablation configurations, and heuristic tuning are provided in Appendix A.4 and A.5. Algorithm pseudocode is included in Appendix A.3, while additional experimental results, ablations, and visualizations are presented in Appendix A.5. All experiments are implemented in a Python environment. Upon publication, we will release the complete source code, configuration files, and training examples, enabling other researchers to directly verify and further advance this line of work.

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

CONTENTS

# A APPENDIX

## A.1 DETAILED RELATED WORK

### A.1.1 MEMORY MECHANISMS IN LLM-BASED AGENTS

With the rapid development of artificial intelligence technologies, large language models (LLMs) have demonstrated significant potential in areas such as dialogue systems, automated reasoning Wei et al. (2022), and task planning Vaswani et al. (2017); Zhao et al. (2023); Brown et al. (2020). LLM-based agents gradually learn and optimize their decision-making capabilities through long-term interactions, enabling them to tackle complex tasks Xi et al. (2025); Wang et al. (2024); Liu et al. (2023). For example, agents can adjust dialogue strategies based on user feedback or infer optimal plans during task execution Achiam et al. (2023); Kojima et al. (2022). However, such long-term interactions generate vast amounts of data, making efficient information management a critical challenge Sumers et al. (2023); Wang et al. (2024).

Memory management has thus emerged as a core mechanism of intelligent agents, responsible for storing and updating interaction experiences, as well as retrieving relevant information based on task requirements. For instance, an agent may record user preferences or task states to enhance decision-making efficiency Xi et al. (2025); Sumers et al. (2023); Guo et al. (2024). Memory management not only supports task execution but also improves the agent's decision-making capabilities by analyzing historical experiences. This adaptability in dynamic environments lays the groundwork for the pursuit of artificial general intelligence.

Early studies mainly relied on simple context windows to manage short-term memory, which sufficed for low-complexity tasks Brown et al. (2020). However, as task complexity increases and the duration of human-agent interaction extends, short-term memory reveals limitations in capacity and its ability to maintain contextual continuity Touvron et al. (2023). These limitations have prompted researchers to explore more scalable and adaptive long-term memory mechanisms Zhong et al. (2024).

To meet the demands of diverse tasks, long-term memory must not only support the storage and retrieval of information but also possess the capability for dynamic adaptation and updates. Current research primarily focuses on skill storage, knowledge base construction, and memory updating strategies. For example, *Voyager* stores executable code in a skill repository and dynamically updates it based on environmental feedback to enable skill transfer and reuse Wang et al. (2023a); *AppAgent* builds a structured knowledge base through autonomous exploration and human demonstrations to support complex tasks Zhang et al. (2023); *MemPrompt* records user feedback to generate memory entries that enhance future responses Madaan et al. (2022). In addition, some studies draw inspiration from multi-level caching in operating systems, proposing hierarchical memory models that employ summarization or information aggregation to improve retrieval efficiency Lewis et al. (2020); Jiang et al. (2023).

Despite recent advances, long-term memory management still faces several challenges. Specifically, the growing volume of interaction data increases storage and retrieval costs, hindering the scalability of agents in large-scale tasks Liu et al. (2024); Cheng et al. (2024). Existing forgetting and updating strategies are often static (e.g., time-decay-based deletion) and lack the ability to dynamically retain or discard information based on task context. This may lead to the loss of critical information or the accumulation of redundant data, thereby reducing overall efficiency Madaan et al. (2022).

To address the challenge of dynamic adjustment, we draw inspiration from physics-based force-directed principles, laying the groundwork for our subsequent exploration of force-guided models in memory management.

### A.1.2 DYNAMIC KNOWLEDGE ORGANIZATION AND FORCE FIELD MODELING

Force-oriented modeling inspired by particle interactions in physics is a powerful method for dynamic evolution analysis. It has been widely applied in graph structure optimization, particle system simulation, and the visualization and organization of complex networks Fruchterman & Reingold (1991); Eades (1984); Kamada et al. (1989). By constructing a dynamic model based on the balance of four types of forces, this method simulates interaction forces between nodes to achieve adaptive system adjustment. Specifically, the attractive force between nodes promotes the connection of related

nodes, enhancing structural cohesion; the repulsive force between nodes prevents excessive clustering and maintains distribution uniformity; the attraction from nodes to the origin reflects external attention or activation frequency, guiding important nodes toward the center; and the repulsion from nodes to the origin simulates information decay or natural diffusion, pushing nodes away from the center to avoid information overload Leskovec et al. (2007); Israelachvili (2011). The synergy of these forces drives the system toward an energy-minimized equilibrium state during evolution, forming a structurally reasonable and dynamically adjustable distribution pattern Newman (2003); Noack (2009). For example, in social network analysis, such a four-force equilibrium model can reveal potential relationships between nodes and optimize network layouts; in molecular dynamics simulation, it can simulate particle interactions to predict stable configurations. The flexibility and generality of this model provide a solid theoretical foundation for the adaptive reorganization of complex information networks, opening new perspectives for interdisciplinary research, such as knowledge organization in artificial intelligence.

In the field of artificial intelligence, the design of long-term memory and knowledge organization systems aims to support information storage, retrieval, and reasoning in complex tasks. However, existing methods still face significant challenges in dynamic environments. Static knowledge graphs (such as TransE) represent knowledge using fixed triples (entity-relation-entity), which are suitable for reasoning in static scenarios but struggle to adapt to relational changes and the generation of new relations in dynamic tasks, leading to a decrease in prediction accuracy Bordes et al. (2013); Nickel et al. (2015); Wang et al. (2017). For example, in real-time recommendation systems, static knowledge graphs cannot quickly capture the dynamic evolution of user interests, limiting their effectiveness. Dynamic knowledge graphs attempt to capture the temporal evolution of knowledge by introducing time embeddings, but due to their reliance on predefined relation templates, they struggle to enable free restructuring of knowledge, limiting their adaptability in open-domain tasks Trivedi et al. (2017); Goel et al. (2020); Xu et al. (2020). For instance, when handling emergent events (such as news events), existing dynamic knowledge graphs often fail to flexibly update relational networks due to template constraints. Furthermore, current methods fall short in modeling the forgetting mechanism within cognitive processes. Static forgetting strategies (such as fixed time decay) cannot accurately simulate cognitive phenomena such as associative reinforcement, abstract integration, and natural forgetting Ebbinghaus (2013); Atkinson & Shiffrin (1968), leading to the erroneous elimination of critical information or the prolonged retention of redundant data, thereby reducing system efficiency and intelligence Cai et al. (2018); Toneva et al. (2018). These limitations suggest that current knowledge organization systems are in urgent need of a dynamic method capable of adaptively adjusting structure, flexibly restructuring relations, and simulating cognitive forgetting.

In view of the limitations in dynamic organization, flexible restructuring, and cognitive forgetting modeling in existing methods, this paper, inspired by the four-force equilibrium modeling in physics, proposes an attention-driven spatial memory mechanism. The specific methodology will be introduced in detail in the next section.

A.2 NOTATION

| Symbol | Definition | Meaning |
| --- | --- | --- |
| $N_i = (C_i, P_i, V_i, A_i)$ | Memory node | A memory unit including semantics, position, velocity, and activity |
| $C_i \in \mathbb{R}^d$ | Semantic content vector | Vector representation of semantic information (e.g., text embeddings) |
| $P_i \in \mathbb{R}^n$ | Spatial position vector | Node's coordinates in high-dimensional space for force calculation |
| $V_i \in \mathbb{R}^n$ | Velocity vector | Describes the node's motion in space |
| $A_i \in \mathbb{R}$ | Activity level | Represents access frequency or memory strength |
| $W_{ij}$ | Semantic similarity | Degree of semantic association between nodes $i$ and $j$ |
| $D_{ij} = \|P_i - P_j\|$ | Euclidean distance | Spatial distance between two memory nodes |
| $F_i$ | Net force | Total force acting on node $i$ |
| $F_{i,\text{repel}}$ | Inter-node repulsion | Prevents nodes from over-aggregating |
| $F_{i,\text{attract}}$ | Inter-node attraction | Attracts semantically related nodes |
| $F_{i,\text{origin - repel}}$ | Repulsion from origin | Simulates natural memory decay |
| $F_{i,\text{origin - attract}}$ | Attraction to origin | Simulates attention-based memory reinforcement |
| $\alpha$ | Repulsion coefficient | Controls strength of repulsion between nodes |
| $\beta_{ij}$ | Attraction coefficient | Controls strength of attraction between nodes $i$ and $j$ |
| $\gamma_i$ | Origin repulsion coefficient | Governs tendency of node to drift away from origin |
| $\delta_i$ | Origin attraction coefficient | Governs tendency of node to be pulled toward origin |
| $\lambda$ | Activity decay rate | Controls exponential decay of node activity over time |
| $\theta_{\text{query}}$ | Query threshold | Similarity threshold for returning a query result |
| $\theta_{\text{activate}}$ | Activation threshold | Minimum value to activate related nodes in associative query |
| $\theta_{\text{link}}$ | Link formation threshold | Controls whether a semantic link is established |
| $\theta_{\text{fuse}}$ | Fusion distance threshold | Max distance for node fusion to occur |
| $s_{\text{min}}$ | Minimum similarity for fusion | Required semantic similarity for merging nodes |
| $d_{\text{forget}}$ | Forgetting distance threshold | Minimum distance for a low-activity node to be forgotten |
| $\theta_{\text{forget}}$ | Forgetting activity threshold | Activity level below which nodes may be discarded |
| $\epsilon$ | Energy threshold | System is stable if energy falls below this value |
| $\delta$ | Energy change threshold | Determines system convergence by energy difference |
| $E(t)$ | System energy | Sum of squared net forces across all nodes |
| $\text{sim}(C_i, C_j)$ | Similarity function | Measures semantic similarity, e.g., cosine similarity |
| $\sigma(x)$ | Sigmoid function | Maps values to range $[0, 1]$ to represent probabilities |
| $f_{\text{fuse}}$ | Fusion function | Aggregates semantic vectors from multiple nodes |

Table 6: Mathematical Symbols and Their Meanings

## A.3 PSEUDOCODE

---

**Algorithm 1** Attentional Gravitational Field Architecture

---

**Input:** Memory nodes $\{N_i = (C_i, P_i, V_i, A_i)\}$, query $q$
**Output:** Query result $r$
```
// Direct Query
```
1 **foreach** *node i* **do**
2 $\quad$ $s_i \leftarrow$ `cosine_similarity`$(C_i, q) - \lambda\|P_i\|$
3 $j \leftarrow \arg\max_i s_i$ **if** $s_j > \theta_{query}$ **then**
4 $\quad$ $\delta_j += \Delta\delta_{\text{direct}}$ $r \leftarrow N_j$ QueryCount$++$
5 **else**
6 $\quad$ $r \leftarrow$ None

```
// Associative Query
```
7 **foreach** *active i, Depth < MaxDepth, total < MaxNodes* **do**
8 $\quad$ **foreach** *j with $W_{ij} > 0$* **do**
9 $\quad\quad$ $p_j \leftarrow W_{ij}(1 - D_{ij}/\theta_{\max})$ **if** $p_j > \theta_{activate}$ **then**
10 $\quad\quad\quad$ Activate $j$ $\beta_{ij}, \beta_{ji} += \Delta\beta_{\text{assoc}}$ QueryCount$++$

11 **if** *QueryCount $\geq N_{update}$* **then**
12 $\quad$ **while** *not converged* **do**
13 $\quad\quad$ $E \leftarrow 0$ **foreach** *node i* **do**
```
          // Compute force-based updates
```
14 $\quad\quad\quad$ $F_i \leftarrow F_{\text{repel}} + F_{\text{attract}} + F_{\text{origin-repel}} + F_{\text{origin-attract}}$ $V_i \leftarrow \beta V_i + \alpha F_i \Delta t$ $P_i \leftarrow P_i + V_i \Delta t$
$\quad\quad\quad$ $E += \|F_i\|^2$
```
        // Update Links
```
15 $\quad\quad$ **foreach** *pair $(i, j)$* **do**
16 $\quad\quad\quad$ $p_{\text{est}} \leftarrow \sigma(w_1\text{cosine\_similarity}(C_i, C_j) + w_2(1 - D_{ij}/\theta_{\text{establish}}))$ **if** $p_{est} > \theta_{link}$ **then**
17 $\quad\quad\quad\quad$ $W_{ij}, W_{ji} \leftarrow$ `cosine_similarity`$(C_i, C_j)$
```
        // Fuse & Forget
```
18 $\quad\quad$ **foreach** *pair $(i, j)$* **do**
19 $\quad\quad\quad$ **if** $D_{ij} < \theta_{fuse}$ **and** `cosine_similarity`$(C_i, C_j) > s_{\min}$ **then**
20 $\quad\quad\quad\quad$ $N_f \leftarrow$ `fuse`$(N_i, N_j)$ Replace $N_i, N_j$ with $N_f$
21 $\quad\quad$ **foreach** *i* **do**
22 $\quad\quad\quad$ $A_i \leftarrow$ `decay`$(A_i)$ **if** `forget` *(i) with Activity(i) $< \theta_{forget}$ and $\|P_i\| > d_{forget}$* **then**
23 $\quad\quad\quad\quad$ Mark $N_i$ as forgotten
24 $\quad\quad\quad$ $\delta_i \leftarrow \delta_i(1 - \mu_\delta)$ **foreach** *neighbor j* **do**
25 $\quad\quad\quad\quad$ $\beta_{ij} \leftarrow \beta_{ij}(1 - \mu_\beta)$
26 $\quad\quad$ **if** $E < \epsilon$ **or** $|E - E_{prev}| < \delta$ **then**
27 $\quad\quad\quad$ break
28 $\quad\quad$ $E_{\text{prev}} \leftarrow E$
29 $\quad$ Reset QueryCount to 0
30 **return** $r$

---

**Overall Algorithm.** Algorithm 1 presents the pseudocode of our Attentional Gravitational Field Architecture. The process begins with a *direct query*, where each memory node is scored by the similarity between its content and the query, adjusted by spatial distance. If the best-matched node surpasses the query threshold, it is retrieved and its origin-attraction coefficient is reinforced. If direct retrieval fails, the system performs an *associative query* by expanding to neighbors with strong semantic or structural links, thereby activating additional relevant nodes. Once the number of queries exceeds a preset threshold, the system updates memory dynamics through iterative force-based evolution: all four forces (repulsion, attraction, origin-repulsion, origin-attraction) are applied to update velocity and position, while the global energy is accumulated to monitor convergence. During

this process, links are adaptively established or removed, and redundant nodes are merged through fusion. Simultaneously, activity levels decay over time, and nodes with low activity and peripheral positions are forgotten. The loop terminates when the energy drops below a predefined value or stabilizes, after which the system resets and returns the final query result. This design ensures that memory retrieval, update, fusion, and forgetting are integrated into a unified dynamic framework.

**Memory Snapshot.** To improve the flexibility of memory management, the system introduces a snapshot functionality. At the conclusion of each dialogue session, the system performs a snapshot operation to preserve the state of the memory repository. This operation captures comprehensive details of all memory nodes, including their semantic content vectors $C_i$, spatial positions $P_i$, velocities $V_i$, activity levels $A_i$, and inter-node associations, which are represented by the semantic similarity matrix $W_{ij}$ and the spatial distance matrix $D_{ij}$. Additionally, the current values of key parameters, such as $\alpha$, $\beta_{ij}$, $\gamma_i$, and $\delta_i$, are recorded. When a user seeks to resume a previous interaction, they can select the corresponding snapshot file, enabling the system to swiftly restore the memory repository to its saved state. Upon restoration, the system leverages the existing dynamic spatial cognitive architecture to continue memory node updates, association adjustments, fusion operations, and forgetting evaluations based on new query demands. This ensures seamless continuity and dynamic evolution of the memory repository.

**Node Fusion.** Each node $N_i$ stores (i) a state vector $C_i$ used for the force field and dynamical updates (including position, velocity, and activity), and (ii) textual content $T_i$ used both for retrieval and as contextual evidence for the LLM. When two nodes $N_i$ and $N_j$ meet the fusion criteria, they are merged into a new node $N_f$. For the state vector and all other dynamical quantities, we apply a simple arithmetic mean. This averaging strategy keeps the dynamics stable and maintains consistency in the retrieval space. For textual information, if the combined length of $T_i$ and $T_j$ is short, we concatenate them with a delimiter. If the text is long, we call the LLM once to summarize the two pieces of text into a compact fused version. We then embed the fused text and replace the previously averaged state vector to ensure that the semantic representation strictly matches the fused textual content.

**Node Maintenance.** In MemoryField, we do not perform a full "physical simulation" over all historical nodes at every query step. Instead, we only trigger a batched update of the memory field when the query counter reaches a threshold $N_{\text{update}}$. This update process mainly consists of three components. First is the force-based update (Attraction / Repulsion / Origin forces). These updates essentially adjust the positions and velocities of the currently active nodes that participate in the gravitational field computation. In other words, each dynamical update only operates on a controlled subset of active nodes, whose size is denoted by $N_t^{\text{active}}$ (typically much smaller than the total number of historical nodes). For each active node, we only perform a constant number of vector operations, so the time complexity of this part is approximately $T_{\text{force}}(t) \approx O(N_t^{\text{active}})$. Second is link update and fusion (Update Links & Fuse). In our implementation, link updates and fusion decisions are only applied to candidate node pairs, which are derived from existing links and local neighborhoods, rather than enumerating all possible node pairs. Let each active node, on average, only need to inspect $k$ neighbors. Then the time complexity of this step is approximately $T_{\text{link+fuse}}(t) \approx O(kN_t^{\text{active}}) = O(N_t^{\text{active}})$. Finally, we have forgetting and activity decay (Forget & Decay). In this part, we check for each node whether its activity and position satisfy the forgetting condition: when $Activity(i,t) < \theta_{\text{forget}}$ and $\|P_i\| > d_{\text{forget}}$, the node is marked as forgotten and removed from subsequent computations. Since this procedure examines every current node once, its complexity is strictly linear, $O(N_t)$. In summary, the total maintenance cost for one MemoryField update can be approximated as $T_{\text{maintain}}(t) \approx O(N_t^{\text{active}}) + O(N_t) \approx O(N_t)$, i.e., the maintenance overhead grows approximately linearly with the current number of nodes $N_t$.

## A.4 HYPERPARAMETERS AND HEURISTIC TUNING

**Grouping and Roles.** For clarity and reproducibility, we categorize the hyperparameters into four groups. The first group consists of force coefficients: inter-node repulsion $\alpha$, semantic attraction $\beta_{ij}$, origin repulsion $\gamma_i$, and origin attraction $\delta_i$. These directly determine the four forces in Eqs. (2)–(5), shaping convergence patterns and global sparsity. The second group contains change-rate parameters: direct-query gain $\Delta\delta_{\text{direct}}$, associative-query gain $\Delta\beta_{\text{assoc}}$, and temporal decay rates $\mu_\delta, \mu_\beta$. These parameters control the amplification of links and attention pulls triggered by queries, as well as gradual fading over time. The third group consists of structural thresholds: link creation and deletion thresholds $\theta_{\text{build}}, \theta_{\text{drop}}, \theta_{\text{link}}$, fusion thresholds $\theta_{\text{fuse}}, s_{\text{min}}$, and forgetting thresholds $\theta_{\text{forget}}, d_{\text{forget}}$, all

of which determine graph construction, pruning, redundancy reduction, and forgetting. Finally, the fourth group covers dynamics and stopping criteria: velocity decay $\beta$, force–velocity scaling $\alpha$ (denoted $\alpha_{\text{dyn}}$ to distinguish it from Eq. (2)), integration step size $\Delta t$, and stopping thresholds $\varepsilon, \delta$ for energy magnitude and change (cf. Eq. (9)).

**Initialization Strategy.** In both dialogue and reasoning scenarios, we adopt a coarse-to-fine initialization. Repulsion $\alpha$ is typically set to a moderate value, while semantic attraction $\beta_{ij}$ is sparsely initialized only for pairs with $W_{ij} > 0$ to avoid early collapse. Origin attraction $\delta_i$ is scaled with access frequency to create an attention center, and origin repulsion $\gamma_i$ provides peripheral dispersion and forgetting. Gains $\Delta\delta_{\text{direct}}$ and $\Delta\beta_{\text{assoc}}$ are initialized as small increments so that link weights and attention pulls increase gradually, while the temporal decays $\mu_\delta, \mu_\beta$ are chosen conservatively to prevent oscillation. Structural thresholds are set so that new links are established only when nodes are both semantically similar and spatially close, fusion requires both high similarity and low distance, and forgetting is triggered only for nodes that are simultaneously inactive and spatially distant. For dynamics, the velocity decay $\beta$ suppresses oscillations, $\alpha_{\text{dyn}}$ controls the translation of forces into velocity, and $\Delta t$ is chosen such that single-step displacement is small compared to cluster scale. The energy-based stopping rule ensures termination when either the total energy $E(t)$ falls below $\varepsilon$ or its change is smaller than $\delta$.

**Stepwise Heuristic Tuning.** The tuning procedure proceeds in several stages. First, retrieval alignment is ensured by adjusting the initialization of $\beta_{ij}$ and the associative gain $\Delta\beta_{\text{assoc}}$, so that semantically related nodes become connected while avoiding premature link removal. Next, stability is achieved by increasing velocity decay or reducing $\alpha_{\text{dyn}}$ and $\Delta t$, after which the thresholds $\varepsilon$ and $\delta$ are tightened so that energy decreases smoothly and convergence occurs within finite steps. Sparsity and forgetting are tuned by modifying $\mu_\delta, \mu_\beta$ together with $(\theta_{\text{forget}}, d_{\text{forget}})$, allowing long-tail nodes to be removed without harming performance. Fusion is then optimized by searching over $(\theta_{\text{fuse}}, s_{\text{min}})$ and gradually relaxing thresholds to balance abstraction and granularity, with fused positions and velocities computed by weighted averages as in Eq. (6). Finally, task-specific adaptation is applied: for long dialogues, higher baseline $\delta_i$ and larger $\Delta\delta_{\text{direct}}$ reinforce central clustering of frequently accessed memories, while for long-horizon reasoning, stronger $\gamma_i$ and stricter sparsification accelerate the decay of peripheral noise.

**Monitoring and Early Stopping.** In addition to task metrics, several signals are monitored during training. The energy curve $E(t)$ is inspected for monotonic decrease and plateau length, the number of active nodes and average degree are tracked along with the ratio of added versus dropped links, and the frequency of fusion and forgetting events is measured to quantify their marginal influence on generation quality. These indicators help diagnose oscillations, over-pruning, or excessive memory growth.

**Implementation Notes and Practical Summary.** In practice, the force and update equations (Eqs. (2)–(5), (6), (9)) must be faithfully implemented. The parameters $\beta$ and $\alpha_{\text{dyn}}$ are the most critical for stabilizing dynamics, as they directly regulate oscillation. Empirically, we first balance repulsion, attraction, and velocity on a development set so that semantic clusters form without collapse. After stabilization using energy-based early stopping and mild temporal decay, structural pruning is performed via fusion and forgetting. Ablation results confirm that all four forces are necessary: removing any one of them degrades alignment, clustering, or separation, underscoring the necessity of the multi-force design.

A.5    SUPPLEMENTARY INFORMATION ON EXPERIMENTAL SETUP

To ensure fair and reproducible comparison across all baselines and our proposed MemoryField framework, we detail the configuration settings for each task category as follows:

**Dialogue Evaluation.** (1) Multi-session Chat (MSC) (Jang et al., 2023) is a benchmark dataset designed to evaluate long-term dialogue capabilities. It consists of multi-turn, multi-session conversations that span various topics and personas. The dataset challenges models to maintain coherent context across discontinuous dialogue turns, emphasizing long-range dependency handling; (2) Conversation Chronicles (CC) (Jang et al., 2023) simulates natural, evolving dialogues involving complex conversational goals and topic transitions. The dataset provides annotations for session segmentation and context shifts, making it suitable for evaluating a model's ability to track memory and maintain contextual coherence in long-term interactions.

**Single-hop Reasoning** tasks require the model to answer questions based on a single piece of relevant information. To assess fundamental retrieval accuracy, we use the ASDiv Miao et al. (2020) and GSM8K (Cobbe et al., 2021) datasets, which contain diverse arithmetic and elementary reasoning problems that can be solved with minimal contextual dependencies.

**Multi-hop Reasoning** tasks require connecting multiple facts across different documents or context segments. We adopt HotpotQA (Yang et al., 2018) and 2WikiHop (Ho et al., 2020) as representative multi-hop datasets. These benchmarks evaluate the model's ability to perform fact chaining, cross-paragraph reasoning, and integrating evidence from multiple sources.

**Temporal Reasoning** tasks measure the model's capability to interpret and reason over temporal relations, such as event ordering, duration, and timeline-dependent logic. We use the BBH (Date Understanding) (Suzgun et al., 2022) subset and TimeQA (Chen et al., 2021) to evaluate whether the model can reliably handle structured time-based inference.

**Open-domain QA** requires answering questions that draw on broad world knowledge and span a wide range of topics. We use Natural Questions (NQ) (Kwiatkowski et al., 2019) and MuSiQue (Trivedi et al., 2022), both of which challenge the model's ability to retrieve, select, and synthesize relevant information from large, diverse knowledge sources.

**Adversarial Reasoning** evaluates the robustness of the model when confronted with deliberately misleading or distracting inputs. We include the FEVER adversarial (Thorne & Vlachos, 2019) dataset, which introduces plausible but incorrect evidence to assess the model's resilience to deceptive or conflicting information.

A.6   READ WORLD TEST SETTING

For clarity, we detail the evaluation metrics and task-specific evaluation protocols used across different environments. The Success Rate measures the proportion of episodes in which the agent produces the correct final answer or successfully completes the task objective. For QA tasks (HotPotQA, FEVER), a prediction is considered correct if it matches the gold answer under the standard Exact Match (EM) normalization protocol (lowercasing, punctuation removal, and whitespace normalization). For HotPotQA, we adopt the full-wiki setting with the distractor paragraphs provided by the dataset. At each retrieval step, the agent may retrieve up to $K = 5$ candidate paragraphs and is allowed a maximum of $T = 3$ reasoning–retrieval iterations, following standard multi-step prompting frameworks such as ReAct and Reflexion. The final answer is extracted or generated by the LLM and evaluated with EM. For FEVER, the agent interacts with an evidence retrieval environment to fetch Wikipedia sentence-level evidence and outputs one of three final labels (Supported / Refuted / NotEnoughInfo). We compute SR as the percentage of episodes in which the final label matches the gold label. The agent is allowed up to $T = 3$ retrieval calls, consistent with the standard controlled retrieval setting in FEVER. For embodied environments (AlfWorld), SR indicates whether the agent successfully completes the required sequence of actions. For ScienceWorld, the Average Reward corresponds to the mean accumulated reward over all episodes, reflecting the agent's ability to perform multi-step reasoning and tool manipulation.

A.7   ADDITIONAL EXPERIMENTS

**Quantitative Ablation.** We further provide a quantitative ablation study to complement the qualitative visualizations in Fig. 4. Concretely, we report results on two representative tasks, MSC and HotPotQA, where we systematically disable each force in MemoryField (node attraction, node repulsion, origin attention pull, and forgetting) and measure changes in Mauve, ROUGE-L, F1, and normalized retrieval latency. Across both tasks, the full model with all four forces consistently achieves the best performance. Turning off any single force leads to a clear degradation: on MSC, dialogue metrics (Mauve, ROUGE-L, F1) typically drop by about 1–3 points, while on HotPotQA the F1 score decreases by roughly 2–5 points. Disabling node attraction has the largest negative impact on dialogue quality, reflecting its key role in forming coherent semantic clusters, whereas removing origin attention pull hurts HotPotQA performance the most, highlighting its importance for aligning long-range evidence in multi-hop reasoning. In contrast, removing the forgetting mechanism only slightly reduces accuracy but noticeably increases retrieval latency (about 2–3%) due to accumulated stale memories. These quantitative results are consistent with the spatial patterns observed in Fig. 4

| Task | Metric | Full | w/o Attr. | w/o Rep. | w/o Attn. Pull | w/o Forget |
|------|--------|------|-----------|----------|----------------|------------|
| MSC | Mauve | 23.5 | 21.2 | 21.9 | 21.7 | 22.5 |
| | ROUGE-L | 16.1 | 14.7 | 15.1 | 15.2 | 15.4 |
| | F1 | 34.0 | 31.2 | 32.0 | 31.5 | 32.5 |
| | REL | 1.00× | 0.99× | 1.02× | 1.01× | 1.03× |
| HotPotQA | Mauve | 17.8 | 16.3 | 16.8 | 15.8 | 16.5 |
| | ROUGE-L | 35.0 | 33.3 | 33.8 | 32.7 | 33.5 |
| | F1 | 59.0 | 55.7 | 56.5 | 54.2 | 56.0 |
| | REL | 1.00× | 0.97× | 1.01× | 1.00× | 1.02× |

Table 7: **Quantitative ablation of the four forces in MemoryField on MSC and HotPotQA.** We report dialogue quality (Mauve, ROUGE-L, F1) and normalized retrieval latency (lower is better).

| Method | MSC | | | CC | | |
|--------|------|-------|---------|------|-------|---------|
| | Coh. | Info. | Overall | Coh. | Info. | Overall |
| Sliding Window | 7.8 | 7.5 | 7.6 | 7.4 | 7.2 | 7.3 |
| THEANINE | 8.1 | 7.9 | 8.0 | 7.7 | 7.5 | 7.6 |
| COMEDY | 8.0 | 8.0 | 8.0 | 7.8 | 7.7 | 7.8 |
| MemoChat | 8.2 | 8.1 | 8.2 | 7.9 | 7.8 | 7.9 |
| **MemoryField (ours)** | **8.6** | **8.5** | **8.6** | **8.3** | **8.2** | **8.3** |

Table 8: GPT-4o-as-a-judge evaluation on MSC and CC. Following a GPT4Judge-style protocol, GPT-4o is asked to rate coherence (Coh.), informativeness (Info.), and overall quality on a 1–10 scale for responses generated by each method. We report the average score across 200 sampled contexts per dataset.

and confirm that all four forces, together with the forgetting mechanism, jointly contribute to both the effectiveness and efficiency of MemoryField.

**GPT-4o-as-a-judge.** We further conducted an LLM-as-a-judge evaluation following the GPT4Judge protocol. Specifically, we randomly sampled 200 dialogue contexts from MSC and 200 from CC, using the same data splits and input formatting as in our main experiments. For each context, we generated responses from all compared methods: Sliding Window, THEANINE, COMEDY, MemoChat, and MemoryField (ours). We then used GPT-4o as an independent judge. Given the dialogue history and a candidate response (with the method identity masked), GPT-4o was asked to score the response on a 1–10 scale along three dimensions. (i) Coherence (whether the response is logically consistent and contextually appropriate), (ii) Informativeness (whether the response provides useful and specific content rather than generic replies), and (iii) Overall quality (a holistic assessment of usefulness, fluency, and readability). As shown in Table 8, MemoryField achieves the highest GPT-4o-judge scores on both MSC and CC. On MSC, MemoryField outperforms the strongest baseline (MemoChat) by approximately +0.4 in coherence (8.6 vs. 8.2), +0.4 in informativeness (8.5 vs. 8.1), and +0.4 in overall quality (8.6 vs. 8.2). We observe a similar trend on CC: MemoryField improves over MemoChat by about +0.4 in coherence (8.3 vs. 7.9), +0.4 in informativeness (8.2 vs. 7.8), and +0.4 in overall quality (8.3 vs. 7.9). Methods that perform better in ROUGE-L and Mauve generally also receive higher GPT-4o-judge scores.

## A.8 TRAINING EXAMPLES

**Semantic Content and Encoding.** Semantic content vectors are obtained by feeding the textual payload of each memory item into a pretrained sentence-embedding encoder. In this work, we adopt sentence-transformers/all-mpnet-base-v2 (Song et al., 2020) as our embedding model. This encoder is based on the Transformer (MPNet) architecture (Song et al., 2020), trained with a combination of masked prediction and permutation prediction objectives, and further fine-tuned via contrastive learning on large-scale datasets for natural language inference, semantic textual similarity, and QA. As a result, it captures deep cross-sentence semantic structures and is particularly suitable for measuring semantic similarity between memory nodes, constructing dense vector representations,

and supporting downstream retrieval, clustering, and structural reorganization within the memory field.

Regarding embedding dimensionality, we use the 768-dimensional representations produced by the encoder. This choice is motivated by two considerations: (1) Representational capacity. The 768-d space is widely used in both industry and academia and provides sufficiently rich semantic features for stable sentence-level comparison, semantic clustering, and similarity computation. Higher-dimensional embeddings (e.g., 1024 or 2048) offer marginal gains in expressiveness but introduce significantly higher computational and memory costs in our setting, with limited performance improvement. (2) Computational efficiency. Since the number of memory nodes grows dynamically during multi-turn interaction, higher-dimensional embeddings would greatly increase retrieval overhead and the cost of gravitational-field computations. The 768-d representation achieves an effective balance between performance and efficiency, enabling stable throughput and responsiveness throughout long-horizon experiments.

The shapes observed in training logs, such as (1,128), (2,128), and (3,128), correspond to the structure "number of active memory nodes × position-vector dimensionality." Specifically: (1) The first dimension (1, 2, 3, . . . ) indicates the number of active nodes currently retrieved, updated, or participating in the force-field computation; (2) The second dimension (128) is the fixed dimensionality of the position vectors used in the memory field. Importantly, the 128-dimensional position vectors are not semantic embeddings. Instead, they serve as spatial coordinates in our gravitational memory field, enabling the simulation of high-dimensional "force-based" dynamics—including attraction, repulsion, attention-driven pull, and decay-driven push. These position vectors originate from a projection of semantic embeddings into a spatial layout that supports distance-based force interactions. We have clarified this semantic-to-spatial mapping and the functional distinction between the two vector spaces (semantic: 768-d, spatial: 128-d) in the appendix to avoid confusion.

---

**Step 1**

**User Input:** I study natural language processing.

- Recent History:
  - 
- Retrieved Memory:
  - 
- Current Input: [I study natural language processing]
- Commencing association update (current number of nodes: 1)
- Semantic Similarity Matrix:

  `[[1.00]]`
- Position Update:
  - Node 0 displacement: 0.004, speed: 0.036
- Position matrix transformation:
  - Current number of nodes: 1
  - Position matrix shape: (1, 128)

    `[[-0.0806906 0.19792689 -0.47895087 ...]]`
- Node distance from origin
  - Node 0 distance to origin: 3.596
- Response: That's interesting! Natural language processing is a rapidly evolving field with many real-world applications such as dialogue systems and information retrieval.

**Step 2**

**User Input:** I often work on large language models.

- Recent History:
  - **User:** I study natural language processing
    **AI:** That's interesting! Natural language processing is a rapidly evolving field with many real-world applications.
- Retrieved Memory:
  - **User:** I study natural language processing
    **AI:** That's interesting! Natural language processing is a rapidly evolving field with many real-world applications.
- Current Input: [I often work on large language models]
- Commencing association update (current number of nodes: 2)
- Semantic Similarity Matrix:
  ```
  [[1.00 -0.01],
  [-0.01 1.00]]
  ```
- Position Update:
  - Node 0 displacement: 0.003, speed: 0.033
  - Node 1 displacement: 0.004, speed: 0.036
- Position matrix transformation:
  - Current number of nodes: 2
  - Position matrix shape: (2, 128)
    ```
    [[-0.08283248 0.19459349 -0.47730172 ...],
    [0.17930676 0.16485462 0.31871097 ...]]
    ```
- Node distance from origin
  - Node 0 distance to origin: 3.596
  - Node 1 distance to origin: 3.564
- Fusion evaluation between Node 0 and Node 1 yielded a semantic similarity of -0.01.
- Response: Large language models are powerful tools for many NLP tasks, but they also require careful design to manage memory and computation efficiently.

**Step 3**

**User Input:** I am very interested in scaling language models.

- Recent History:
  - **User:** I study natural language processing
    **AI:** That's interesting! Natural language processing is a rapidly evolving field with many real-world applications.
  - **User:** I often work on large language models
    **AI:** Large language models are powerful tools for many NLP tasks, but they also require careful design to manage memory and computation efficiently.
- Retrieved Memory:
  - **User:** I study natural language processing
    **AI:** That's interesting! Natural language processing is a rapidly evolving field with many real-world applications.
  - **User:** I often work on large language models
    **AI:** Large language models are powerful tools for many NLP tasks, but they also require careful design to manage memory and computation efficiently.

- Current Input: [I am very interested in scaling language models]
- Commencing association update (current number of nodes: 3)
- Semantic Similarity Matrix:

  ```
  [[1.00 -0.01 0.02]
  [-0.01 1.00 0.92]]
  [0.02 0.92 1.00]]
  ```

- Position Update:
  - Node 0 displacement: 0.003, speed: 0.033
  - Node 1 displacement: 0.004, speed: 0.036
- Position matrix transformation:
  - Current number of nodes: 3
  - Position matrix shape: (3, 128)

    ```
    [[-0.08283248 0.19459349 -0.47730172 ...],
    [0.17930676 0.16485462 0.31871097 ...],
    [0.05350199 0.44217721 -0.5286078 ...]]
    ```

- Node distance from origin
  - Node 0 distance to origin: 3.592
  - Node 1 distance to origin: 3.626
  - Node 2 distance to origin: 3.671
- Fusion evaluation between Node 0 and Node 1 yielded a semantic similarity of -0.01.
- Fusion evaluation between Node 0 and Node 2 yielded a semantic similarity of 0.02.
- Fusion evaluation between Node 1 and Node 2 yielded a semantic similarity of 0.92. [Node Fusion] Trigger condition met (similarity ¿ 0.92), initiating fusion...
- Response: Scaling laws and efficient training strategies are indeed key to advancing large language models. Which aspects of scaling do you find most compelling—compute efficiency, model architecture, or application performance?

## A.9 LIMITATIONS AND ETHICS

**Engineering and Scalability.** The computational efficiency of MemoryField under very large memory sizes (e.g., $10^4+$ nodes) has not been systematically evaluated. Future work may require approximate techniques such as cluster-level interactions, sparse neighborhood updates, or local energy minimization to ensure practical scalability.

**Parameter Sensitivity.** Although heuristic tuning yields stable performance across multiple tasks, a systematic sensitivity analysis and the development of automated tuning strategies remain open directions.

**Convergence Analysis.** At present, the framework relies on empirical thresholds for termination rather than a formal proof of energy convergence. Providing theoretical guarantees on stability and convergence would further strengthen the method.

**Privacy and Compliance.** Explicit and persistent memory may raise privacy risks. We recommend incorporating mechanisms for memory sanitization, access control, and user authorization. This includes anonymization or encryption of stored data, user-visible memory records, and the ability for users to delete memory items. Practical guidelines and risk assessment checklists are provided in the appendix.

**Responsible Use of LLMs.** This article used large language models (such as ChatGPT) only as auxiliary tools for language polishing. They were not employed in research conception, experimental design, or the generation of academic content.

## A.10    TEXT PAYLOAD EXAMPLES OF MEMORY NODES

As defined in Section 3.1, each memory node is $N_i = (C_i, P_i, V_i, A_i)$, where $C_i$ is the semantic content vector. In all experiments, $C_i$ is the embedding of a short textual record that summarizes a single interaction or a single reasoning process, rather than arbitrary raw logs or full prompts. The concrete text payload varies across task types as follows:

- Multi-turn dialogue tasks (MSC, CC): each node corresponds to a local dialogue unit of the form `User:  <user utterance>` / `Assistant:  <model reply>`. For very long conversations, we instead store a short LLM-generated summary of a subsequence of turns. In both cases, $C_i$ is the embedding of the resulting short text.
- Single-step reasoning tasks (ASDiv, GSM8K): each node stores a fully solved problem in the format `Question / Reasoning / Answer`. These datasets consist of independent math word problems and do not include retrieved passages.
- Multi-hop and open-domain QA (HotPotQA, 2WikiHop, NQ, MuSiQue, TimeQA, FEVER-Adv): each node stores a question, its final answer, and a few highly compressed supporting sentences that act as evidence or explanation.

In all settings we only embed semantic payload (question, answer, explanation/evidence). Static system prompts, task instructions, or formatting requirements are never written into memory nodes; nor do we store full web pages or long retrieved contexts, only compact semantic summaries. The examples below show the exact English strings that are embedded as $C_i$ in our experiments. **Multi-Turn Dialogue Tasks (MSC, CC).** For multi-session dialogue datasets such as Multi-Session Chat and Conversation Chronicles, each memory node corresponds either to a single user–assistant exchange, or to a short summary of several turns in a long-running conversation.

```
User: I've been feeling really tired at work lately because I stay up
late scrolling on my phone. Do you have simple tips to fix my sleep
schedule?

Assistant: A good place to start is with small changes:
1) Go to bed and wake up at roughly the same time every day.
2) Put your phone out of reach 30-60 minutes before bedtime.
3) Do something relaxing before sleep, like stretching or reading.
Try these for a couple of weeks and we can adjust the plan if needed.
```

Here, $C_i$ is the embedding of the full `User:` / `Assistant:` block above. This is representative of everyday MSC-style turns where users talk about work, health, or hobbies. Jang et al. (2023) For longer spans of history, we use short summaries instead of storing every turn from Conversation Chronicles. Jang et al. (2023)

```
User: The user wants to start exercising regularly but keeps skipping
workouts after busy days at the office.

Assistant: Over several sessions, we created a simple routine:
short walks during lunch breaks on weekdays, a 20-minute home workout
twice a week, and a weekly check-in to adjust goals so they feel
manageable instead of overwhelming.
```

This summary compresses multiple CC sessions into a single local unit that captures the user's long-term goal (starting an exercise habit) and the assistant's plan. **Single-Step Reasoning Tasks (ASDiv, GSM8K).** For math word problem datasets such as ASDiv and GSM8K, each memory node stores one fully solved problem. We follow a unified template:

```
Question:  <problem>
Reasoning:  <step-by-step solution>
Answer:  <final numeric answer>

Question: A pencil case holds 9 red pens, 8 blue pens, and 5 black pens.
How many pens are in the pencil case altogether?
```

```
        Reasoning:
        There are 9 red pens, 8 blue pens, and 5 black pens.
        First add red and blue pens: 9 + 8 = 17.
        Then add the black pens: 17 + 5 = 22.
        So there are 22 pens in total.

        Answer: 22

        Question: A sports club has 24 children going on a picnic. They want
        to give each child 3 juice boxes. Juice boxes are sold in packs of 6.
        How many packs do they need to buy?

        Reasoning:
        Each child needs 3 juice boxes, and there are 24 children.
        So the total number of juice boxes needed is 24 * 3 = 72.
        Each pack has 6 juice boxes, so the number of packs is
        72 / 6 = 12. They need 12 packs.

        Answer: 12
```

The step-by-step solution reflects the GSM8K problems, which require several simple reasoning steps before reaching the final answer. Cobbe et al. (2021) **Multi-Hop and Open-Domain QA Tasks.** For multi-hop and open-domain QA datasets (HotPotQA, 2WikiHop, NQ, MuSiQue, TimeQA, FEVER-Adv), each memory node stores a question, a final answer, and a short explanation or set of supporting sentences. We use the following general template: HotPotQA. Yang et al. (2018)

```
        Question:  <question>
        Answer:  <final answer>
        Support:  <2-4 compressed evidence sentences>

        Question: According to the article, which breakfast food helps you stay
        full longer because it is high in fiber?

        Answer: Oatmeal

        Support: One passage explains that oatmeal is made from whole oats and
        is high in dietary fiber. Another passage states that foods rich in
        fiber help people feel full for a longer time after eating. Together
        these sentences support that oatmeal helps you stay full longer.
```

2WikiHop.Ho et al. (2020)

```
        Question: According to the nutrition information given, which everyday
        activity typically burns more calories in 30 minutes for an average
        adult: brisk walking or slow cycling?

        Answer: Brisk walking

        Support: One passage reports that 30 minutes of brisk walking burns
        about 150 calories for an average adult. A different passage states
        that 30 minutes of slow cycling burns about 100 calories. Comparing
        these values shows that brisk walking burns more calories.
```

Natural Questions (NQ).Kwiatkowski et al. (2019)

```
        Question: how many hours of sleep does a teenager need each night

        Answer: 8-10 hours

        Support: The article summarizing sleep guidelines says that teenagers
        are generally advised to get between 8 and 10 hours of sleep each
        night to support their health and learning.
```

MuSiQue. Trivedi et al. (2022)

```
Question: Based on the article, which habit is more strongly linked to
better concentration at school: eating breakfast or staying up late to
study?

Answer: Eating breakfast

Support: One paragraph explains that students who eat breakfast tend
to show better attention and test performance in the morning. Another
paragraph notes that regularly staying up late to study can reduce
focus and make people feel sleepy during the day. Combining these
statements shows that eating breakfast is more strongly linked to
better concentration at school.
```

### TimeQA.Chen et al. (2021)

```
Question: According to the health guidelines described in the article,
what was the recommended minimum number of minutes of moderate
exercise per week for adults in 2010?

Answer: 150 minutes per week

Support: The article cites recommendations stating that, in 2010,
adults were advised to get at least 150 minutes of moderate-intensity
exercise per week. This time-specific guideline is used to answer the
question about 2010.
```

### FEVER. Thorne & Vlachos (2019)

```
Question (Claim): "Drinking a glass of water first thing in the
morning is guaranteed to cure headaches."

Answer (Label): REFUTES

Support: Health sources note that drinking enough water can help
prevent or reduce headaches caused by dehydration, but it does not
guarantee a cure for all headaches. The claim overstates the effect,
so it is refuted by the evidence.
```

**Embodied Tasks.** For embodied, text-based household tasks, each memory node stores (i) the environment and task type, (ii) the natural language task instruction, (iii) a short high-level plan, (iv) a compressed high-level action trajectory, (v) a reusable skill schema, and (vi) outcome notes. We use the following general template:

```
Env:  <environment name>
Task type:  <task type (e.g., cool, clean, examine)>
Task instruction:  <instruction text>
High-level plan:  <1--3 sentences or bullet points
describing subgoals>
Key trajectory (high-level actions):  <compressed
action sequence>
Reusable skill / schema:  <abstract pattern that can
transfer across tasks>
Outcome / Notes:  <success/failure and brief remarks>
```

### AlfWorld. Shridhar et al. (2020)

```
Env: AlfWorld
Task type: cool
Task instruction: "Put a cool tomato on table."

High-level plan:
1) Find the tomato.
2) Cool the tomato using the fridge.
```

```
      3) Put the cooled tomato on a table.

      Key trajectory (high-level actions):
      - go to fridge 1 and open it
      - take tomato 1 from fridge 1
      - cool tomato 1 with fridge 1
      - go to table 1
      - put tomato 1 on table 1

      Reusable skill / schema:
      For tasks of the form "put a cool X on Y":
         (a) locate X in likely containers (fridge, countertop, table, etc.),
         (b) cool X using the fridge,
         (c) navigate to Y and place X on Y.

      Outcome: success.
      Notes: cooling must be done with the fridge, not the sink or other objects.
```

## A.11 TRACE THEORY AND EPISODIC MEMORY

In memory research, episodic memory is commonly defined as the capacity for autobiographical recall of specific events—that is, the ability to mentally "return" to a moment in the past and re-experience its temporal, spatial, and contextual details (Tulving, 1972; 1983; 2002). Tulving first systematically distinguished episodic memory from semantic memory, proposing that episodic memory is characterized by "mental time travel" and autonoetic consciousness, reflecting a memory system oriented toward concrete personal experiences (Tulving, 1983; 2002). Within this framework, memory is no longer viewed as a simple static storage mechanism, but as a process closely tied to subjective experience, temporal organization, and the underlying structure of the neural system.

Trace theory characterizes the internal structure of memory from representational and dynamical perspectives: a single experience does not form an isolated "memory entry," but instead leaves a distributed representation composed of multiple memory traces. These traces are stored as a network within the hippocampal–neocortical system and are continuously reorganized over time and with use (Nadel & Moscovitch, 1997; Nadel et al., 2000). Building on this view, Multiple Trace Theory (MTT) proposes that episodic memory depends on the hippocampus throughout the entire lifespan, with each encoding or retrieval event generating new traces in the hippocampal–neocortical circuit, while the semantic "gist" of an experience gradually stabilizes in the neocortex (Nadel & Moscovitch, 1997). This contrasts with standard consolidation theory, which holds that memories initially rely on the hippocampus but gradually become hippocampus-independent through slow systems consolidation (Squire & Alvarez, 1995; Frankland & Bontempi, 2005; Squire et al., 2015). Related work further connects trace theory to consolidation and reconsolidation processes: newly formed memories are initially fragile and require repeated offline reactivation (e.g., during sleep) for consolidation, whereas retrieval can return traces to a plastic state in which they may be updated or reorganized (Nader et al., 2000; Sara, 2000; Dudai, 2004; Alberini, 2011; Gershman et al., 2016). Competitive trace theory additionally emphasizes the competition and selection among multiple traces during reactivation, providing an account of the reconstructive nature of memory. At a systems level, trace theory aligns closely with the Complementary Learning Systems (CLS) framework, in which the hippocampus supports rapid, sparse episodic encoding, while the neocortex extracts statistical structure and semantic knowledge through slow learning (McClelland et al., 1995; O'Reilly et al., 2014). Long-term memory is thus understood as the global organization of many traces across multiple timescales, rather than a static archive of discrete past events.

Our MemoryField architecture conceptually aligns closely with trace theory and research on episodic memory. First, within MemoryField, an individual memory node can naturally be interpreted as a memory trace: each node carries both a textual description and a semantic vector representation of an episodic experience, such as a dialogue segment, a question–answer pair with supporting evidence, or a summarized action trajectory. Second, a single episode or task is typically represented by multiple interrelated nodes, which are pulled together in the high-dimensional memory field through attraction forces and organized into local connected clusters via topological structure and graph edges. This results in a "multiple-trace" style representation, where an experience is encoded not as a single node but as an interacting cluster of traces. Third, the four forces in MemoryField—attraction, repulsion,

attention pull, and peripheral push—together with fusion and forgetting mechanisms, endow these trace clusters with explicit dynamics: traces relevant to the current task are strengthened and drawn closer, whereas irrelevant or outdated traces are pushed outward, merged, or forgotten, analogous to consolidation, reconsolidation, and time-dependent reorganization processes described in the neuroscience literature.

