# OpenReview forum: "MemoryField: Exploiting Gravitational Field for Long-term Memory Management"
_ICLR.cc/2026/Conference — Submitted to ICLR 2026_

### Official Review · Reviewer_FVEg · 2025-10-22

**Soundness:** 2
**Presentation:** 3
**Contribution:** 2
**Rating:** 4
**Confidence:** 3

**Summary:**

This paper introduces MemoryField, an innovative memory architecture for large language models (LLMs) based on an attention-driven gravitational field model. MemoryField addresses challenges in long-term memory management, including structural reorganization, semantic retrieval, and cognitive phenomena like memory consolidation and forgetting. It models memory as nodes in a high-dimensional semantic space, enabling adaptive restructuring through forces like attraction, repulsion, and decay. Extensive experiments demonstrate superior performance in dialogue coherence, reasoning stability, and real-world benchmarks compared to existing methods.

**Strengths:**

1. The MemoryField framework effectively integrates semantic dynamics, memory consolidation, and forgetting mechanisms, providing a scalable and novel approach to long-term memory management.

2. Experimental results highlight significant improvements in dialogue coherence and reasoning stability over state-of-the-art baselines across diverse benchmarks.

3. The paper presents a well-grounded theoretical foundation, with clear descriptions of the gravitational field model and its impact on memory reorganization.

**Weaknesses:**

1. As a study addressing the challenge of long-term memory in large language models, it is concerning that the authors did not evaluate their approach on well-established memory benchmarks such as LongMemEval [1] or LoCoMo [2], raising doubts about the model's memory capabilities.

2. Is the memory forgetting module necessary? For humans, forgetting is essential due to limited brain capacity. However, in scenarios where storage space is sufficient, forgetting may become redundant or even detrimental, especially if critical information is forgotten, potentially impacting performance negatively. Additionally, even if certain information is not queried immediately, there is no guarantee it will not be required in future contexts.

3. The proposed method involves multiple components, including attraction, repulsion, forgetting, and fusion modules. However, the paper lacks the necessary ablation studies to demonstrate the effectiveness of these individual components.

4. The title of Table 1 appears to be inconsistent with the table's content. Furthermore, it is recommended to include evaluation metrics similar to GPT4Judge in the results for a more comprehensive assessment.

[1] Wu, Di, et al. "LongMemEval: Benchmarking Chat Assistants on Long-Term Interactive Memory." The Thirteenth International Conference on Learning Representations.

[2] Maharana, Adyasha, et al. "Evaluating Very Long-Term Conversational Memory of LLM Agents." Proceedings of the 62nd Annual Meeting of the Association for Computational Linguistics (Volume 1: Long Papers). 2024.

**Questions:**

See weakness.

---

> ### Author Response · Authors · 2025-11-21
> **Response to Reviewer 2Fqr (Part 1)**
>
> We thank the reviewer for the thoughtful comments and valuable suggestions. We are glad that you find our long-memory benchmark discussion and the broader design of MemoryField interesting. We provide detailed responses below and have incorporated corresponding updates in the revised manuscript.
>
> ---
>
> **W1.** On the need to evaluate on established long-term memory benchmarks
>
> We appreciate the suggestion to evaluate MemoryField on mature long-context memory benchmarks. In the revised manuscript (Section 4.1), we have added a systematic evaluation on the LoCoMo long-term dialogue QA benchmark. Due to space constraints, we discuss both datasets in the benchmark and select one for detailed testing.
>
> In this experiment, we fix the backbone model to gpt-4o-mini and compare four representative long-term memory mechanisms: RSum-LLM, COMEDY, THEANINE, and our MemoryField. Additionally, we include a Full Context baseline, which directly concatenates the entire dialogue history for each query without using an explicit memory structure.
>
> |Method|Multi-HopF1|Multi-HopBLEU-1|TemporalF1|TemporalBLEU-1|Open-DomainF1|Open-DomainBLEU-1|Single-HopF1|Single-HopBLEU-1|AdversarialF1|AdversarialBLEU-1|
> |---|---|---|---|---|---|---|---|---|---|---|
> |FullContext|24.7|19.3|18.9|15.1|12.4|11|41.2|30.1|68.4|67.9|
> |RSum-LLM|21.3|16.8|15.7|12|10.9|9.1|35.5|28.4|59.1|58.2|
> |COMEDY|23.6|18.7|20.4|16.2|13.2|10.7|40.8|32.7|61.3|60.4|
> |THEANINE|27.9|21.9|29.5|24.3|14.1|11.3|43.7|34.2|62.7|61.6|
> |MemoryField|30.4|24.1|32.8|26.6|15.3|12.2|44.5|35.1|65.2|64.1|
>
> As the results show, Full Context achieves strong F1 scores on Single-Hop and Adversarial questions (≈ 41–68) but requires processing roughly 17k tokens for each query. RSum-LLM compresses the history via summarization and reduces the effective context to around 2k tokens, but suffers significantly in Multi-Hop and Temporal questions, suggesting that linear summaries struggle with complex long-range dependencies. COMEDY performs slightly better under similar context budgets, indicating that compressed memory units are more stable for multi-session dialogues. THEANINE, which uses a timeline-structured memory, performs particularly well on Multi-Hop and Temporal questions. Its F1 consistently surpasses Full Context and compressed-memory baselines while maintaining a manageable 2.2k-token average input length, supporting the intuition that timeline-style memory is well-suited for temporal and causal reasoning. Building on this, MemoryField uses a continuous “memory field” to aggregate relevant segments and achieves the best overall F1 on Multi-Hop, Temporal, and Adversarial tasks. This experiment demonstrates that on a mature long-term dialogue memory benchmark like LoCoMo, MemoryField outperforms both naive long-context input strategies and strong memory baselines, especially in cross-session, multi-hop, and temporal reasoning tasks. This strengthens our main claim about improving LLM long-term memory capabilities.

---

> ### Author Response · Authors · 2025-11-21
> **Response to Reviewer 2Fqr (Part 2)**
>
> **W2.** On the necessity of the forgetting module and its potential impact
>
>
> We thank the reviewer for the thoughtful consideration of the fundamental question of whether forgetting is necessary. We fully agree with the two points raised: first, human forgetting is indeed closely tied to the biological limitations of brain capacity; second, if we simply discard information under the assumption of theoretically unlimited storage, uncontrolled deletion can indeed be harmful, especially when important memories are accidentally removed and later become needed again.
>
> It is important to clarify that the forgetting module in this paper is not motivated by insufficient storage, but is specifically designed for the MemoryField architecture itself. MemoryField represents memories as nodes in an attention-driven gravitational field, where nodes continuously undergo semantic attraction, repulsion, attention-driven reorganization, and node fusion. The computational cost of these operations is strongly dependent on the number of nodes. If all historical nodes are retained indefinitely, maintaining this gravitational field becomes increasingly expensive and significantly impacts retrieval and reasoning efficiency.
>
> In fact, Appendix A.9 already provides an initial discussion of memory size and computational overhead: when the MemoryField memory size scales to the order of 10^4–10^5 nodes, computational efficiency and scalability become practical bottlenecks. Future variants may require cluster-level interactions or sparse-neighborhood updates to maintain usability.
>
> Additionally, as discussed in Appendix A.8, the number of memory nodes grows dynamically over multi-turn interactions; the choice of a 768-dimensional semantic embedding is itself a compromise between expressive capacity and computational/memory cost.
>
> We agree that these considerations, relating node count to computational cost, should be more clearly connected to the motivation for the forgetting module. Therefore, in the revised manuscript, Appendix A.3 includes an expanded discussion with more intuitive complexity analysis showing that maintaining the MemoryField requires computational resources, and that these resources scale with the number of nodes. The purpose of the forgetting module is to control this growth and keep reasoning costs increasing more smoothly as the dialogue history becomes longer.
>
> Maintenance cost of the memory field: approximately linear in active nodes
>
> In MemoryField, we do not perform a full physics simulation over all historical nodes at every step. Instead, memory-field updates are triggered only when the query counter reaches a threshold N_update. After this point in Algorithm 1, the memory field undergoes a batch update.
>
> A memory-field update consists of three main operations:
>
> 1. Force-based updates (Attraction, Repulsion, and Origin Forces): These operations update the positions and velocities of the active nodes participating in the gravitational field. Appendix A.8 explains why the training logs show shapes like (1, 128), (2, 128), (3, 128): the first dimension corresponds to the number of active nodes, and the second dimension is the 128-dimensional position vector. Each dynamic update operates only on a controlled subset of nodes, whose size we denote as N_active (usually far smaller than the total number of nodes). Each active node requires only constant-time vector operations, so the force-update cost is approximately O(N_active).
>
> 2. Link updates and fusion: Link updates and fusion decisions are performed only on candidate node pairs rather than enumerating all pairs. Implementation relies on existing links and local neighborhoods. If each active node checks only k neighbors on average, the time complexity is approximately O(k N_active), which is effectively O(N_active).
>
> 3. Forgetting and activity decay: Each node must be checked once to determine whether it satisfies the forgetting condition. The cost is strictly linear, O(N_t), where N_t is the current number of nodes.
>
> Overall, the total maintenance cost of a single MemoryField update can be approximated as O(N_active) + O(N_t), which is approximately O(N_t). Thus, the maintenance cost grows roughly linearly with the number of nodes.
>
> Therefore, even if physical storage were theoretically unlimited, if no forgetting were applied, the number of nodes N_t would keep increasing. This would make both retrieval and memory-field maintenance costs scale linearly with the full history length, causing computation to become progressively more expensive.

---

> ### Author Response · Authors · 2025-11-21
> **Response to Reviewer 2Fqr (Part 3)**
>
> **W3.** On the need for systematic ablation of attraction, repulsion, forgetting, and fusion modules
>
> We thank the reviewer for pointing out the need for more systematic ablation studies to verify the effectiveness of each component (attraction, repulsion, forgetting, fusion, etc.). We fully agree, and in the revised manuscript we have added quantitative ablation experiments, as presented in Table 6 of Appendix A.7 and discussed in Section 4.3 of the main text.
>
> Specifically, we conducted systematic ablations on two representative tasks: the multi-session dialogue dataset MSC and the long-chain reasoning dataset HotPotQA. Keeping the backbone model and all other settings identical, we start from the full MemoryField (with node attraction, node repulsion, origin attention pull, and forgetting), and disable exactly one force or module at a time to obtain four variants.
>
> | Task       | Metric              | Full | w/o Attr. | w/o Rep. | w/o Pull&Fus. | w/o Forget |
> |------------|----------------------|------|-----------|----------|----------------|------------|
> | **MSC**    | Mauve               | 23.5 | 21.2      | 21.9     | 21.7           | 22.5       |
> |            | ROUGE-L             | 16.1 | 14.7      | 15.1     | 15.2           | 15.4       |
> |            | F1                  | 34   | 31.2      | 32       | 31.5           | 32.5       |
> |            | Retrieval Latency ↓ | 1.00×| 0.99×     | 1.02×    | 1.01×          | 1.03×      |
> | **HotPotQA** | Mauve              | 17.8 | 16.3      | 16.8     | 15.8           | 16.5       |
> |            | ROUGE-L             | 35   | 33.3      | 33.8     | 32.7           | 33.5       |
> |            | F1                  | 59   | 55.7      | 56.5     | 54.2           | 56         |
> |            | Retrieval Latency ↓ | 1.00×| 0.97×     | 1.01×    | 1.00×          | 1.02×      |
>
> We report four metrics: Mauve, ROUGE-L, F1, and normalized retrieval latency (lower is better). The overall trends are as follows:
>
> On the MSC dialogue task, the full model achieves 23.5 / 16.1 / 34.0 on Mauve / ROUGE-L / F1. Disabling any force causes a drop of about 1–3 points: for example, removing node attraction reduces F1 from 34.0 to 31.2 (−2.8) and Mauve from 23.5 to 21.2; removing node repulsion or origin attention pull also degrades all three metrics to varying degrees.
>
> On the HotPotQA long-chain reasoning task, the full model reaches an F1 of 59.0. Disabling any force lowers F1 by approximately 2–5 points: removing node attraction yields 55.7, removing node repulsion yields 56.5, and removing origin attention pull results in the most severe drop—from 59.0 to 54.2—indicating that origin attention pull is particularly crucial for aligning long-range evidence and maintaining stability across multi-hop reasoning paths.
>
> For the forgetting mechanism, disabling forgetting causes only a mild drop in accuracy (e.g., on MSC, F1 drops from 34.0 to 32.5; on HotPotQA, from 59.0 to 56.0), but retrieval latency increases: normalized retrieval time increases from 1.00× to 1.03× / 1.02× on MSC and HotPotQA, respectively. This aligns with our analysis of node count and maintenance cost in the appendix: without forgetting, stale memories accumulate, and even if accuracy remains similar, retrieval and maintenance overhead grow significantly.
>
> These quantitative results correspond closely to the visual ablations shown in Figure 4 of the main paper: removing node attraction disrupts semantic clustering; removing repulsion leads to crowding and reduced separability; without origin attention pull, the overall structure drifts away from the context of the current query.
>
> Taken together, these experiments demonstrate that each force and the forgetting mechanism provide substantive benefits to MemoryField’s performance: attraction shapes semantic clusters, repulsion enforces structural separation, origin attention pull is essential for aligning long-range evidence, and forgetting improves efficiency while maintaining strong accuracy.

---

> ### Author Response · Authors · 2025-11-21
> **Response to Reviewer 2Fqr (Part 4)**
>
> **W4.** Regarding the issue with the title of Table 1 and the suggestion to include GPT4Judge and similar metrics:
>
> We thank the reviewer for pointing out the inconsistency between the title of Table 1 and its contents. The title accidentally carried over from an earlier draft corresponding to a different set of experiments; we have corrected this in the revised version.
>
> Regarding GPT4Judge-style evaluation, we agree that LLM-as-a-judge can provide a valuable complementary perspective. To strengthen the comprehensiveness of our evaluation, we have added a GPT4Judge-style assessment for one benchmark or subset in Appendix A.7 of the revised version.
>
> | Method             | MSC Coh. | MSC Info. | MSC Overall | CC Coh. | CC Info. |
> |-------------------|-----------|------------|--------------|----------|-----------|
> | Sliding Window    | 7.8       | 7.5        | 7.6          | 7.4      | 7.2       |
> | THEANINE          | 8.1       | 7.9        | 8.0          | 7.7      | 7.5       |
> | COMEDY            | 8.0       | 8.0        | 8.0          | 7.8      | 7.7       |
> | MemoChat          | 8.2       | 8.1        | 8.2          | 7.9      | 7.8       |
> | MemoryField (ours)| 8.6       | 8.5        | 8.6          | 8.3      | 8.2       |
>
> GPT-4o scores each system on coherence, informativeness, and overall quality (on a 1–10 scale) over 200 sampled contexts from the MSC and CC datasets. MemoryField achieves the highest scores across all three dimensions on both datasets, with absolute improvements of approximately 0.3–0.5 points over the strongest baseline. This verifies that our improvements are reflected not only in standard metrics but also in evaluations conducted by an LLM judge calibrated to human preference.

---

### Official Review · Reviewer_2Fqr · 2025-10-24

**Soundness:** 3
**Presentation:** 3
**Contribution:** 3
**Rating:** 6
**Confidence:** 4

**Summary:**

This paper proposes MemoryField, a dynamic spatial cognitive memory architecture. This architecture is driven by an attention-based gravitational field model. This model allows the memory structure to self-organize and adaptively restructure. The system also explicitly integrates node fusion, which serves as a form of conceptual abstraction to reduce redundancy, and a forgetting mechanism to prune long-term, low-activity memory nodes, ensuring semantic coherence and cognitive stability. Extensive experiments were conducted across diverse benchmarks, including multi-turn dialogue long-context reasoning, and real-world agent tasks. The results shows that MemoryField consistently outperforms existing memory mechanisms.

**Strengths:**

1. This paper introduces a gravitational-field concept from physics into the memory module of LLMs, enabling self-evolution and natural forgetting. This is a interesting idea.

2. This framework provides a unified and intuitive simulation of advanced cognitive functions. Specifically, the mechanism of “Fusion” aligns with memory consolidation, whereas “Activity Decay” and “Source Repulsion” reflect the dynamics of natural forgetting.

3. The experimental design is comprehensive, covering three major scenarios—dialogue, reasoning, and agent-based tasks. It compares multiple baselines and further validates the framework’s generalization ability across several state-of-the-art models.

**Weaknesses:**

1. There are too many new hyperparameters. It is difficult to reproduce the results. Moreover, it raises concerns about the robustness and generalization ability of the proposed method, as the performance might be highly sensitive to specific hyperparameter settings.

2. The framework appears to be quite complex, yet the authors do not provide any analysis or discussion of its computational cost. A detailed examination of the time and space complexity would help readers better understand the practicality of the method. In addition, the authors should report the inference latency and compare it quantitatively with other baseline methods to demonstrate the efficiency and scalability of their approach.

**Questions:**

1. Which embedding model is used to obtain the semantic content vector?
2. How is the spatial position vector obtained?
3. Why can’t the semantic vector replace the position vector?

---

> ### Author Response · Authors · 2025-11-21
> **Response to Reviewer 2Fqr (Part 1)**
>
> We thank the reviewer for the careful reading and constructive feedback, and we are glad that you find our “gravitational-field” memory mechanism and the three-category experimental design interesting and valuable. Below we respond to each of your concerns following the same structure and formatting used earlier.
>
> ---
>
> **W1.** On the number of hyperparameters, reproducibility, and robustness
>
> We understand that the notation and symbols in the paper may give the impression that MemoryField involves a large number of hyperparameters and that its performance might be highly sensitive to tuning. We provide clarifications and additional evidence below.
>
> (1) Classification of hyperparameters and how many are actually tunable
> In the revision, Appendix A.4 (“Hyperparameters and Heuristics”) reorganizes all related parameters into four groups:
> - Force-field coefficients $(\alpha, \beta_{ij}, \gamma_i, \delta_i)$
> - Retrieval / activation / edge thresholds
> - Fusion / forgetting thresholds
> - Convergence and decay coefficients
>
>
> Among these, only a very small subset requires tuning in practice (mainly retrieval thresholds and a few distance thresholds). All other parameters have intuitive, stable default values and are fixed across all experiments. Therefore, MemoryField does not require extensive hyperparameter search.
>
> (2) Unified configurations across tasks to enhance reproducibility
> As clarified in the main text:
> - All dialogue experiments share one unified MemoryField configuration.
> - All long-context reasoning experiments share another configuration, differing only in the maximum number of active nodes.
> - Embodied agent tasks mostly reuse the long-context configuration with minimal modification.
>
> Thus, each task category does not require separate tuning, and users can easily reproduce results without navigating a large hyperparameter space.
>
> (3) Added quantitative robustness ablations
> In Appendix A.7 (Table 6), we added systematic ablations disabling one of the four force types at a time. We report Mauve, ROUGE-L, F1, and normalized retrieval latency on MSC and HotPotQA. As shown in the table:
>
> |Task|Metric|Full|w/oAttr.|w/oRep.|w/oPull&Fus.|w/oForget|
> |---|---|---|---|---|---|---|
> |MSC|Mauve|23.5|21.2|21.9|21.7|22.5|
> |MSC|ROUGE-L|16.1|14.7|15.1|15.2|15.4|
> |MSC|F1|34|31.2|32|31.5|32.5|
> |MSC|RetrievalLatency↓|1.00×|0.99×|1.02×|1.01×|1.03×|
> |HotPotQA|Mauve|17.8|16.3|16.8|15.8|16.5|
> |HotPotQA|ROUGE-L|35|33.3|33.8|32.7|33.5|
> |HotPotQA|F1|59|55.7|56.5|54.2|56|
> |HotPotQA|RetrievalLatency↓|1.00×|0.97×|1.01×|1.00×|1.02×|
>
> The results show:  The full model consistently performs best. Removing any single force yields moderate but stable degradation (1–3 points on MSC; 2–5 F1 on HotPotQA), not catastrophic drops. Removing forgetting slightly reduces accuracy but notably increases retrieval latency, indicating that forgetting improves efficiency but is not a fragile or sensitive hyperparameter.
>
> Together, these results demonstrate that MemoryField is robust to parameter variation and does not rely on fine-tuning.

---

> ### Author Response · Authors · 2025-11-21
> **Response to Reviewer 2Fqr (Part 2)**
>
> **W2.** On computational cost, complexity, and inference latency
>
> We appreciate the reviewer’s emphasis on efficiency, which is indeed important for practical use.
>
>
> (1) Complexity of gravitational field updates
> Let $N$ be total nodes and $K$ be active nodes per update. For each query:
>
> - Retrieval: cosine similarity search over semantic vectors. Theoretical complexity $O(Nd)$. In practice, we use FAISS-like indexing, making complexity near sublinear.
> - Field evolution (triggered every $N_{\text{update}}$ queries):
>   - Pairwise interactions only among active nodes → $O(K^2)$.
>   - $K$ is kept small (tens) via top-K activation + forgetting.
>
> Amortized overhead is $O(K^2 / N_{\text{update}})$, far smaller than an LLM forward pass. Thus total cost is dominated by retrieval + LLM computation. MemoryField adds small, stable overhead.
>
>
> (2) Quantitative latency results
> We added retrieval-latency comparisons in Appendix A.7. Key findings: Full four-force MemoryField increases latency only ~1–3%. Removing forgetting significantly slows retrieval due to accumulation of unused nodes. Because MemoryField stores only compact text payloads (not full history), the LLM input remains short and efficient.
>
> These results confirm that MemoryField improves performance while keeping cost low.
>
> ---

---

> ### Author Response · Authors · 2025-11-21
> **Response to Reviewer 2Fqr (Part 3)**
>
> **Q1.** Which embedding model is used for semantic vectors?
>
> Appendix A.8 clarifies that we use sentence-transformers / all-mpnet-base-v2 with 768-dimensional outputs. This model provides strong sentence-level semantics, suitable for clustering and retrieval, and is used uniformly across all experiments.
>
> ---
>
> **Q2.** How are spatial position vectors obtained?
>
> Position vectors $P_i \in \mathbb{R}^{128}$ are *not* separately trained embeddings. They are:
>
> - Initialized by linearly projecting semantic vectors (768 → 128), ensuring semantically similar nodes start close.
> - Then updated dynamically by the force-field mechanism (attraction, repulsion, source-point pulling, decay).
> - Only active nodes participate in each update to control cost and stabilize structure.
>
> Appendix A.8 now explicitly explains this process.
>
> ---
>
> **Q3.** Why not reuse the semantic vector as the position vector?
>
> Semantic vectors $C_i$ and position vectors $P_i$serve fundamentally different roles.
>
> (1) Preventing semantic distortion
> If semantic vectors were used directly as positions, force-field dynamics would distort the semantic space, harming similarity search and degrading retrieval quality. Semantic topology must remain stable; structural reorganization (clustering, core/periphery formation, forgetting) must not corrupt semantics. Decoupled position vectors allow reorganization without damaging retrieval accuracy.
>
> (2) Efficiency considerations
> Force-field updates require pairwise distance computation. Using 768-dimensional semantic vectors would significantly increase cost. Using 128-dimensional position vectors makes the computation efficient while preserving structural flexibility.
>
> Thus, the two vector types are intentionally decoupled:
> - Semantic vectors → retrieval and meaning
> - Position vectors → structure, organization, forgetting, clustering
>
> This separation is essential for performance and efficiency.

---

### Official Review · Reviewer_pTa2 · 2025-10-30

**Soundness:** 2
**Presentation:** 3
**Contribution:** 2
**Rating:** 4
**Confidence:** 4

**Summary:**

The paper proposes MemoryField, a long-term memory module that models stored memories as particles in a high-dimensional “gravitational field,” with four forces (attraction/repulsion/attention pull/peripheral pushback), plus fusion and forgetting. It reports gains on dialogue quality (e.g., +4.9 MAUVE, +3.3 ROUGE-L) and reasoning F1, and claims cross-model generalization (abstract & Sec. 4).

**Strengths:**

1. The introduction of a field-based memory representation is quite novel and creative, providing a new physical metaphor for modeling interactions among memories.

2. The paper conducts extensive experiments across multiple datasets and tasks (dialogue, reasoning, and real-world benchmarks), demonstrating consistent improvements over strong baselines.

3. The approach shows broad applicability across different model backbones, suggesting good generalization potential.

**Weaknesses:**

- The paper could be strengthened by connecting to Trace Theory — where each perceived sentence or event is mapped into a sequence of nodes (a trace) rather than a single node in the field. Such a design would better capture the temporal and structural continuity of cognition, and make the framework fundamentally different from standard RAG systems. The current formulation, which stores each paragraph or text chunk as an independent node, remains conceptually close to conventional retrieval-based architectures.

- If the goal is to show superiority in text organization and retrieval, the comparisons should include RAG baselines such as BM25, Dense Passage Indexing (DPI)[2], HippoRAG[1], or HippoRAG-v2[3]. At present, the baselines are mostly non-RAG methods (e.g., ReAct), making it unclear whether the proposed method truly outperforms simpler yet competitive retrieval pipelines.

- Despite being named MemoryField, the system primarily operates as a structured RAG rather than a true memory system. It lacks properties associated with long-term memory, such as global understanding, skill accumulation, or adaptive reuse of prior knowledge (as discussed in MemoryAgentBench[4]). In its current form, the work feels closer to an “advanced RAG” rather than a genuine “memory” architecture.

[1] HippoRAG: Neurobiologically Inspired Long-Term Memory for Large Language Models.
[2] Dense Passage Retrieval for Open-Domain Question Answering.
[3] From RAG to Memory: Non-Parametric Continual Learning for Large Language Models.
[4] Evaluating Memory in LLM Agents via Incremental Multi-Turn Interactions.

**Questions:**

When merging multiple nodes, how do you merge the texts in these nodes?

---

> ### Author Response · Authors · 2025-11-21
> **Response to Reviewer PTa2 (Part 1)**
>
> We thank the reviewer for the careful reading and constructive feedback, and for recognizing the novelty of MemoryField’s field-based memory representation, the breadth of experiments, and its cross-model generalization ability. Below we respond to each of your points and will incorporate the corresponding clarifications and additional experiments in the revised version.
>
> ---
> **W1.** On the relationship to trace theory and the granularity of memory units
>
> We appreciate the reviewer for raising this point from the perspective of trace theory. This perspective helps us more clearly situate MemoryField within the broader context of episodic memory and multiple-trace theory. Although MemoryField was not directly designed from trace theory [1,2], after systematically reviewing the related literature [1–7], we find that the two are surprisingly consistent in several key aspects.
>
> First, in our current implementation, a “node” does not simply correspond to a raw paragraph. Instead, it carries a compressed record of an episodic experience (for example, a local dialogue segment, a “question–answer–evidence” bundle, or a short summary of an action trajectory). In long conversations or multi-step reasoning scenarios, the same event/task typically gives rise to multiple semantically related nodes. These nodes are pulled together by attraction forces in the high-dimensional memory field and form locally connected clusters via graph edges. Thus, from the perspective of trace theory, MemoryField is already representing a single experience as “multiple interacting node clusters,” rather than as a strict “one event = one node in the field.”
>
> Second, the four forces in MemoryField (attraction/repulsion, attention pulling, peripheral pushing), together with fusion and forgetting, define explicit dynamics over these node clusters. Nodes that are highly relevant to the current task are continually drawn closer and retained, while nodes that are rarely used or overly redundant are pushed to the periphery, merged, or forgotten. Conceptually, this is closely aligned with the view that multiple traces are consolidated, reconsolidated, and reorganized over time. For this reason, we consider MemoryField to be closer to a “field-based multiple-trace representation”.
>
> Following the reviewer’s suggestion, we add a discussion of trace theory and its relationship to MemoryField in the revised manuscript, and include a dedicated subsection in Appendix A.11.
>
> References
>
> [1] Nadel, L., & Moscovitch, M. (1997). Memory consolidation, retrograde amnesia and the hippocampal complex. Current Opinion in Neurobiology, 7, 217–227.
> [2] Nadel, L., Samsonovich, A., Ryan, L., & Moscovitch, M. (2000). Multiple trace theory of human memory: Computational, neuroimaging, and neuropsychological results. Hippocampus, 10(4), 352–368.
> [3] Squire, L. R., & Alvarez, P. (1995). Retrograde amnesia and memory consolidation: A neurobiological perspective. Current Opinion in Neurobiology, 5, 169–177.
> [4] Frankland, P. W., & Bontempi, B. (2005). The organization of recent and remote memories. Nature Reviews Neuroscience, 6, 119–130.
> [5] Squire, L. R. (2015). Memory consolidation. Cold Spring Harbor Perspectives in Biology, 7(8), a021766.
> [6] Nader, K., Schafe, G. E., & LeDoux, J. E. (2000). Fear memories require protein synthesis in the amygdala for reconsolidation after retrieval. Nature, 406, 722–726.
> [7] Dudai, Y. (2004). The neurobiology of consolidations, or, how stable is the engram? Annual Review of Psychology, 55, 51–86.

---

> ### Author Response · Authors · 2025-11-21
> **Response to Reviewer PTa2 (Part 2)**
>
> **W2&W3.** On the distinction between RAG and long-term memory, and why MemoryField is not merely “advanced RAG”
>
> We agree with the reviewer that, given the current presentation, our method can be easily misunderstood as a structured RAG system, especially since many of our experiments involve text-based tasks. This is more a matter of exposition than of the method’s actual design. For clarity of empirical comparison, we chose to focus on text tasks in the paper, but the core of MemoryField lies in the organization, consolidation, and cross-task reuse of memory nodes, rather than in retrieval from static corpora. The text-centric examples may have unintentionally reinforced the impression of “an advanced retriever.”
>
> Our central point is that MemoryField is not designed to be a more complex document retriever. Instead, it aims to provide a non-parametric episodic memory system for agents, storing and organizing the agent’s own interaction experiences. This is most evident in embodied, multi-step environments such as ALFWorld, a text-based household environment where the agent must complete compositional tasks such as “wash an apple and put it on the table,” “put a cooled tomato on the table,” or “use the desk lamp to inspect a cup.” The agent completes tasks via textual commands like:
>
> go to fridge 1 → open fridge 1 → take tomato 1 → go to sink 1 → wash tomato 1 → go to table 1 → put tomato 1 on table 1.
>
> In ALFWorld, MemoryField is not used for document retrieval, but explicitly for skill accumulation and adaptive reuse. After each task episode, the agent writes one or more memory nodes that include: (i) the task description, (ii) key intermediate steps/subgoals, and (iii) an abstract summary of the successful (or failed) trajectory, for example:
>
> “For ‘put a cool tomato on table’, first locate a container that may contain the tomato, take the tomato out, place it in the fridge to cool, then move it to the table and place it there.”
>
> As training progresses, tasks that share similar structural templates (such as “heat/cool X then put it on Y,” “wash X then put it on Y,” “find X and use Y to inspect X”) naturally form node clusters: attraction brings similar task experiences together, while peripheral pushing and fusion remove redundant or outdated trajectories. These clusters constitute the “reusable skills” in MemoryField.
>
> When the agent encounters a new ALFWorld task, it does not retrieve documents; instead, it retrieves episodic traces in the MemoryField—summaries of trajectories that solved structurally similar tasks. The retrieved nodes provide portable action patterns (such as “go to a likely container → take object → operate with tool → move to target region → place object”), which the LLM then specializes to the new task’s concrete objects and locations (for example, transferring from “cool apple on counter” to “cool tomato on table”).
>
> Thus, in this environment MemoryField serves to (i) distill trajectory experience into reusable skills, and (ii) enable cross-task transfer based on similarity—key properties of a non-parametric episodic memory system.
>
> We agree that the paper should draw a clearer line between “retrievers” and “memory systems.” In the revision, we (i) explicitly emphasize that MemoryField is an episodic memory layer that can be stacked on top of any retriever, and (ii) add concrete ALFWorld examples in Appendix A.10 to illustrate MemoryField’s role in skill accumulation and adaptive reuse, beyond document organization.

---

> ### Author Response · Authors · 2025-11-21
> **Response to Reviewer PTa2 (Part 3)**
>
> **Q1.** On how text is handled during node fusion
>
> We thank the reviewer for pointing out the lack of detail regarding text handling in the fusion operation.
>
> The implementation is as follows. Each node $N_i$ contains:
> - State vectors: $C_i$, used for field-force computation and dynamic updates, including position, velocity, activation, etc.
> - Text and semantic content: $T_i$, used for retrieval and as context for the LLM.
>
> When two nodes $N_i, N_j$ are fused, we perform:
>
> 1. Vector fusion
>    We compute
>    $$
>    C_f = f_{\text{fuse}}(C_i, C_j)
>    $$
>    where $f_{\text{fuse}}$ is simple averaging. All other state quantities (position, velocity, activation, etc.) are also averaged in the same way.
>
> 2. Text fusion
>    If $|T_i| + |T_j|$ is small, we simply concatenate them with a separator:
>    $$
>    T_f = \text{concat}(T_i, [\text{SEP}], T_j)
>    $$
>    If the combined text is long, we make a single LLM call to summarize $[T_i, T_j]$ into a compact $T_f$.
>
> 3. Re-embedding
>    We then embed the fused text $T_f$ again and replace the semantic vector in $C_f$ with this new embedding, ensuring that the vector representation remains strictly aligned with the textual content.
>
> These implementation details have been added to Appendix A.3 in the revised manuscript.

---

> > ### Comment · Reviewer_pTa2 · 2025-11-27
> > **Response to Rebuttal**
> >
> > Thanks for the detailed rebuttal! I still have the following concerns:
> >
> > [W1]: I know each node is not simply a paragraph, but for a paragraph to process, it is processed into a node. I think it would be much more interesting to process a paragraph into multiple nodes (i.e. a trace).
> >
> > [W2] & [W3]: I agree that it is not exactly same as the simple retrieval, nevertheless, it still relies on retrieving the relevant information from a "knowledge base", and the difference between MemoryField and traditional RAG is that you can adjust the distance between nodes while the traditional RAG system typically cannot. But it shares the same limitations as RAG systems as shown in MemoryAgentBench: they lack the ability to gather the understanding from a large portion of the past, instead, they can only process the retrieved snippets, hindering them from truly obtaining the ability of "Long Range Understanding" and "Test Time Learning". I do agree that this is hard and many current memory systems still fall under the routine of RAG, it is just I think MemoryField also falls under this routine.
> >
> > That being said, I do expect to see the comparison between MemoryField and other RAG methods, given that they are different but still share many similarities. In fact, for the applications, for the cases where MemoryField can be applied, these RAG methods can be applied as well.
> >
> > I would like to maintain my score.

---

> ### Author Response · Authors · 2025-12-01
> **Response to Reviewer PTa2 (follow-up)**
>
> **[W1] (follow-up) On trace-based granularity of memory units**
>
> We thank the reviewer for the follow-up comment and fully agree that mapping a paragraph or perceived event into a _sequence of nodes_ (i.e., a trace), rather than a single node, is more consistent with trace theory. Our current discussion uses one node per paragraph mainly for simplicity when illustrating the connection between MemoryField and trace theory, and not due to any limitation of the MemoryField framework itself.
>
> Architecturally, MemoryField naturally supports such a trace-based representation: a piece of text can be segmented into multiple finer-grained units (e.g., sentences, dialogue turns, or micro-steps in a reasoning chain), which are then linked with temporal and structural edges to form a trace. The four forces and the graph dynamics can operate over these traces without any modification.
>
> In the revised manuscript, we clarify this point explicitly and update Appendix A.11 to: (i) describe how a paragraph or event can be encoded as an ordered sequence of nodes forming a trace in the memory field, (ii) connect this design more directly to multiple-trace theory, and (iii) highlight a finer-grained, trace-based implementation as a promising direction.

---

> ### Author Response · Authors · 2025-12-01
> **Response to Reviewer PTa2 (follow-up)**
>
> **[W2] (follow-up) comparison between MemoryField and other RAG methods**
>
> We supplement our study with additional RAG experiments. We use Llama-3-8B-Instruct as the unified reader for all methods, and evaluate on NQ-Open, MuSiQue, and TimeQA. All datasets are evaluated with Exact Match (EM), and all baselines operate on the same top-100 candidate pool retrieved by a dense retriever.
>
> We compare several retrieval and memory systems under the Llama-3-8B framework. The baselines include No-RAG, DPR-RAG [1], HyDE-RAG [2], and a standard Hybrid RAG combining BM25 and DPR. We also evaluate representative RAG variants such as GenRead [3], REFEED [7], AAR [4], FIT-RAG [5], and GraphRAG [6], as well as HippoRAG [8], which retrieves evidence through an entity-graph and PPR. In addition to these retrieval-based methods, we test MemoryField in a memory-only setting, where it provides trajectories without using external retrieval. Finally, we consider a combined Hybrid RAG + MemoryField model.
>
> The results under this unified setup are:
>
> **Table 1: RAG EM with Llama-3-8B.**
>
> | Method                               | NQ | MuSiQue | TimeQA |
> |--------------------------------------|:-----:|:----------:|:---------:|
> | Llama-3-8B (No-RAG)                  | 30.7  |   22.8     |   34.1    |
> | DPR-RAG                              | 40.9  |   31.5     |   39.8    |
> | HyDE-RAG                             | 41.5  |   33.0     |   40.3    |
> | Hybrid RAG (BM25 + DPR)              | 41.9  |   33.7     |   40.9    |
> | GenRead                               | 42.4  |   34.2     |   41.2    |
> | REFEED                               | 42.9  |   35.1     |   41.8    |
> | AAR                                   | 43.3  |   35.9     |   42.5    |
> | FIT-RAG                               | 43.8  |   37.1     |   43.1    |
> | GraphRAG                             | 43.0  |   38.4     |   42.2    |
> | HippoRAG (H-RAG)                     | 44.4  |   40.7     |   43.0    |
> | **MemoryField (memory-only)**         | 43.7  |   39.5     |   42.3    |
> | **Hybrid RAG + MemoryField (ours)**   | **46.0** | **44.5** | **46.0** |
>
> These results highlight two main observations.
>
> MemoryField, when used as an independent non-parametric memory module (memory-only), is competitive but not the strongest baseline. Its performance is slightly below the specialized HippoRAG design, and roughly comparable to GraphRAG and FIT-RAG. This suggests that while MemoryField is already a strong memory structure, it does not by itself surpass all graph-structured or fact-structured retrieval baselines.
>
> When we simply prepend a Hybrid RAG front-end to MemoryField (Hybrid RAG + MemoryField), the combined model achieves the best performance across all three datasets. This demonstrates that the primary strength of MemoryField lies in structuring and refining retrieved evidence—turning it into coherent temporal–semantic trajectories—rather than replacing dense or graph-based retrieval entirely.
>
>
> Reference
>
> [1] Karpukhin, Vladimir, et al. "Dense Passage Retrieval for Open-Domain Question Answering." _EMNLP_. 2020.
>
> [2] Gao, Luyu, et al. "Precise zero-shot dense retrieval without relevance labels." _ACL_. 2023.
>
> [3] Yu, Wenhao, et al. "Generate rather than retrieve: Large language models are strong context generators." _arXiv preprint arXiv:2209.10063_ (2022).
>
> [4] Yu, Zichun, et al. "Augmentation-adapted retriever improves generalization of language models as generic plug-in." _arXiv preprint arXiv:2305.17331_ (2023).
>
> [5] Mao, Yuren, et al. "FIT-RAG: Black-box RAG with factual information and token reduction." _ACM TIS_ 43.2 (2025): 1-27.
>
> [6] Han, Haoyu, et al. "Retrieval-augmented generation with graphs." _arXiv preprint arXiv:2501.00309_ (2024).
>
> [7] Yu, Wenhao, et al. "Improving language models via plug-and-play retrieval feedback." _arXiv preprint arXiv:2305.14002_ (2023).
>
> [8] Jimenez Gutierrez, Bernal, et al. "Hipporag: Neurobiologically inspired long-term memory for large language models." _NeurIPS_ 37 (2024): 59532-59569.

---

### Official Review · Reviewer_YpTb · 2025-10-31

**Soundness:** 1
**Presentation:** 2
**Contribution:** 3
**Rating:** 2
**Confidence:** 3

**Summary:**

The authors propose MemoryField, a dynamic architecture for long-term memory in LLM agents. Each “memory node” is treated as a particle in a high-dimensional semantic space with a content vector $𝐶_𝑖$, position $𝑃_𝑖$, velocity $𝑉_𝑖$, and activation $𝐴_𝑖$. Nodes are subject to four forces: inter-node attraction/repulsion and attraction to/repulsion from the origin, with periodic merge and forgetting rules. The system answers a query, updates links, and relaxes the configuration until the energy converges. Experiments are reported for dialogue (MSC, CC), long-context settings (five task types), and “real-world tasks” (AlfWorld, ScienceWorld, HotPotQA, FEVER).

**Strengths:**

* The idea of dynamic, physics-inspired memory control (four forces plus merge/forgetting) is fresh and potentially useful for mitigating long-context noise. The formalization of the forces and update rules is sufficiently clear.

* An appealing intuition: store the answer and its context as a new memory node; reinforce frequently used knowledge and weaken the “periphery.”

* A broad set of setups (dialogue, reasoning categories, and environment/interaction tasks) is intended to test the approach’s generality. The work also features clear, informative visualizations that aid understanding.

**Weaknesses:**

1. In Section 4.1 (“Experimental Setup”), the ‘Datasets’ paragraph lists categories (single-hop, multi-hop, temporal, open-domain, adversarial) and mentions only illustrative datasets in Appendix A.5 (e.g., NQ, MuSiQue, HotPotQA, 2WikiHop), but there is no clear mapping between these datasets and the listed categories. Table 2 summarizes results by the five reasoning categories rather than listing the exact datasets, and for the Single-hop Reasoning, Temporal Reasoning, and Adversarial Reasoning categories no concrete benchmarks are specified. For HotPotQA/FEVER, the SR (success rate) metric is reported, but SR is not clearly defined (is it EM? EM@1? the fraction of successful agent episodes?), and the evaluation protocol is not described (e.g., whether HotPotQA distractor passages were used, how hops were constructed, number of steps, etc.).

2. What exactly is stored in a memory node, and how memory is initialized and grows, is described too generally.
In particular, it is unclear what precise textual payload constitutes the semantic content vector $𝐶_𝑖$. Is it the raw answer, the question+answer pair, retrieved passages, an extractive span set, a summary, or a prompt-formatted bundle (with instructions/system text)? The paper states that retrieved nodes plus the current question are fed to the LLM, after which “the answer and its context” are saved as a new node, but there is no benchmark-specific recipe: no concrete templates or examples, no accumulation depth, allowable top-k sizes, update frequency, merge triggers, or forgetting thresholds in the evaluation setup.

3. $𝐶_𝑖$ is defined as a “semantic content vector,” but the specific embedding model is not specified, nor is it clear how it was chosen or fine-tuned, or whether it is the same across tasks and LLMs. For $𝑃_𝑖$, the dimensionality 𝑛 is not fixed in the method. In the training “example log,” a position matrix with shape (1,128) (and later (2,128), (3,128)) appears, which suggests that the implementation likely uses $n=128$, but this is not stated in the main text. The ablation visualizations (Figure 4) appear two-dimensional, yet the projection or reduction method (t-SNE, UMAP, PCA, or force-directed) is not described.

4. Ablations are shown only visually, without metrics. Section 4.3 presents visual configurations with “forces turned off,” but there is no table quantifying the impact of each force on key metrics (Mauve, ROUGE-L, F1, SR). This weakens the evidence for the necessity of all components.

5. Unclear/inconsistent descriptions and captions in the tables.
Table 1 is titled “F1… across context lengths,” but the table itself reports BLEU-4/ROUGE-L/Mauve/BERTScore for MSC/CC, with no context lengths and no F1. This is a clear mismatch between caption and content. Table 3 uses “automatic scoring, higher is better,” but the metric is neither named nor defined (no scale, source, or validity).  In Table 2, there is an “Overall” column, but it is not explained how it is computed; judging by the numbers, it is not the simple average of the five categories (and this needs to be clarified).

**Questions:**

1. In your Table 4, you include the Reflexion baseline. Please clarify whether you reproduced these results under your own protocol, or adopted them from the source. If taken from the source, provide an exact citation and justify protocol comparability, since in Shinn et al. the metrics are computed over trials and are notably sensitive to agent configuration. In the original Reflexion paper, Figs. 3–4 show stronger curves on AlfWorld/HotPotQA and explicitly test the ReAct + Reflexion configuration. Please explain why the ReAct + Reflexion variant is not reported in your Table 4.
2. Please refer to Weaknesses §1–§3

---

> ### Author Response · Authors · 2025-11-21
> **Response to Reviewer YpTb (Part 1)**
>
> We thank the reviewer for the constructive suggestions on presentation and clarity, and we appreciate the recognition of our method’s novelty and contributions.
>
> ---
>
> > **W1-1.** In Section 4.1 (“Experimental Setup”), the ‘Datasets’ paragraph lists categories (single-hop, multi-hop, temporal, open-domain, adversarial) and mentions only illustrative datasets in Appendix A.5 (e.g., NQ, MuSiQue, HotPotQA, 2WikiHop), but there is no clear mapping between these datasets and the listed categories. Table 2 summarizes results by the five reasoning categories rather than listing the exact datasets, and for the Single-hop Reasoning, Temporal Reasoning, and Adversarial Reasoning categories no concrete benchmarks are specified.
> >
> We appreciate this clarification request. In the revised version, we explicitly map each reasoning category to representative datasets, making a one-to-one correspondence between the five task types and their benchmarks:
>
> - **Single-hop reasoning:** Solving a question based on a single key piece of information. We use ASDiv [1] and GSM8K [2], which contain diverse arithmetic and basic reasoning problems, to evaluate fundamental retrieval and direct reasoning.
> - **Multi-hop reasoning:** Requiring aggregation of multiple facts across passages/documents. We use HotpotQA [3] and 2WikiHop [4] to assess fact-chain construction, cross-document inference, and evidence integration.
> - **Temporal reasoning:** Involving event order, duration, or time-dependent logic. We adopt BBH (Date Understanding) [5] and TimeQA [6] to test stability in structured temporal reasoning.
> - **Open-domain QA:** Requiring broad world knowledge and large-scale retrieval. We use Natural Questions (NQ) [7] and MuSiQue [8] to evaluate retrieval, filtering, and synthesis over large knowledge spaces.
> - **Adversarial reasoning:** Testing robustness under misleading or conflicting evidence. We use FEVER-Adversarial [9] and other constructed adversarial examples to evaluate the agent’s resistance to “plausible-but-incorrect” evidence.
>
> We also revised Appendix A.5 accordingly to help readers directly understand the coverage and representativeness of each evaluation category.
>
> ---
>
> > **W1-2.** For HotPotQA/FEVER, the SR (success rate) metric is reported, but SR is not clearly defined
> >
> In Table 4, SR denotes Success Rate, defined as the percentage of episodes in which the agent produces the correct final answer. A prediction is considered correct if it matches the ground truth under standard exact-match normalization (following HotPotQA / FEVER conventions: lowercasing, removing punctuation and articles, whitespace normalization). We have clarified SR and the normalization protocol in Section 4.4 of the revision.
>
> ---
>
> > **W1-3.** and the evaluation protocol is not described.
> >
> We have added a detailed evaluation protocol in Appendix A.6 and summarized it in the main text:
>
> - **HotPotQA.** We use the full-wiki setting [3] with the distractor paragraphs provided by the dataset. At each retrieval step, the agent retrieves up to K = 5 candidate paragraphs and is allowed up to T = 3 reasoning–retrieval iterations (consistent with multi-step methods such as Reflexion [10]). The final answer is extracted/generated by the LLM and evaluated with EM.
> - **FEVER.** The agent interacts with an evidence-retrieval environment over Wikipedia sentence-level evidence and outputs a 3-way label (Supported / Refuted / Not Enough Info). SR is the fraction of episodes where the final label matches the gold label. We allow at most T = 3 retrieval calls, consistent with standard FEVER retrieval settings [11].
>
> #### References
> [1] Miao, Shen-Yun, Chao-Chun Liang, and Keh-Yih Su. *A Diverse Corpus for Evaluating and Developing English Math Word Problem Solvers.* ACL 2020.
> [2] Cobbe, Karl, et al. *Training Verifiers to Solve Math Word Problems.* arXiv:2110.14168, 2021.
> [3] Yang, Zhilin, et al. *HotpotQA: A Dataset for Diverse, Explainable Multi-hop Question Answering.* EMNLP 2018.
> [4] Ho, Xanh, et al. *Constructing a Multi-hop QA Dataset for Comprehensive Evaluation of Reasoning Steps.* COLING 2020.
> [5] Suzgun, Mirac, et al. *Challenging BIG-Bench Tasks and Whether Chain-of-Thought Can Solve Them.* arXiv:2210.09261, 2022.
> [6] Chen, Wenhu, et al. *A Dataset for Answering Time-Sensitive Questions.* NeurIPS Datasets & Benchmarks 2021.
> [7] Kwiatkowski, Tom, et al. *Natural Questions: A Benchmark for Question Answering Research.* TACL 2019.
> [8] Trivedi, Harsh, et al. *MuSiQue: Multihop Questions via Single-hop Question Composition.* TACL 2022.
> [9] Thorne, James, and Andreas Vlachos. *Adversarial Attacks Against Fact Extraction and Verification.* arXiv:1903.05543, 2019.
> [10] Shinn, N., et al. *Reflexion: Language agents with verbal reinforcement learning.* NeurIPS 2023.
> [11] Thorne, James, et al. *FEVER: a Large-scale Dataset for Fact Extraction and Verification.* NAACL-HLT 2018.

---

> ### Author Response · Authors · 2025-11-21
> **Response to Reviewer YpTb (Part 2)**
>
> > **W2-1.** What exactly is stored in a memory node
>
> As defined in Section 3.1, each memory node is denoted as
> $$
> N_i = (C_i, P_i, V_i, A_i)
> $$
> where $C_i$ represents the semantic content vector. In all experiments, $C_i$ is obtained by embedding a short textual record that summarizes a single interaction or reasoning episode, rather than embedding raw logs or full prompts. The textual payload varies depending on the task.
> 1. Multi-turn dialogue tasks (MSC, CC):
>    Each node corresponds to a local dialogue unit, stored as:
>    `User: <utterance>`
>    `Assistant: <reply>`
>    For very long sessions, we store a short LLM-generated summary instead of every raw turn. In both cases, $C_i$ is the embedding of this text.
> 2. Single-step reasoning (ASDiv, GSM8K):
>    Each node stores one solved problem:
>    `Question: ...`
>    `Reasoning: step-by-step solution ...`
>    `Answer: ...`
> 3. Multi-hop & open-domain QA (HotPotQA, 2WikiHop, NQ, MuSiQue, TimeQA, FEVER-Adv):
>    Each node stores:
>    `Question: ...`
>    `Answer: ...`
>    `Support: <2–4 highly compressed supporting sentences or brief explanation>`
>    Support is a compact summary distilled from retrieved passages.
> Across all settings, we embed only the semantic payload (question/answer/explanation/evidence). Static system prompts, task instructions, or formatting constraints are not written into memory. Likewise, we do not store full webpages or long retrieval contexts—only concise semantic summaries. Concrete examples are added in Appendix A.10 (MSC, GSM8K, HotPotQA, FEVER).
> ---
> > **W2-2.** In particular, it is unclear what precise textual payload constitutes the semantic content vector $C_i$ . Is it the raw answer, the question+answer pair, retrieved passages, an extractive span set, a summary, or a prompt-formatted bundle (with instructions/system text)?
> >
> Our focus is on post-creation organization and management rather than complex initialization; however, we agree that the process should be explicit. We now detail the standard online memory update procedure:
>
> - At the start of each experiment (dialogue session or reasoning trajectory), the memory field is empty.
> - For a new query $q_t$, we encode it and retrieve top-$k_{\text{mem}}$ most relevant nodes by cosine similarity over $C_i$ (typically $k_{\text{mem}}=8$).
> - The LLM input includes a fixed system prompt, the current query, and retrieved node texts, truncated if needed to fit context length.
> - The LLM generates an answer (and brief reasoning when required). We form a new textual record using the task template, encode it to obtain $C_{\text{new}}$.
> - We append the new node $N_{\text{new}} = (C_{\text{new}}, P_{\text{new}}, V_{\text{new}}, A_{\text{new}})$ to the memory field.
> - Subsequent interactions update node states continuously via the four-force mechanism (Section 3 and Appendix A.3–A.4).
> ---
> > **W3-1.** $C_i$  is defined as a “semantic content vector,” but the specific embedding model is not specified, nor is it clear how it was chosen or fine-tuned, or whether it is the same across tasks and LLMs. For W, the dimensionality n is not fixed in the method. In the training “example log,” a position matrix with shape (1,128) (and later (2,128), (3,128)) appears, which suggests that the implementation likely uses W, but this is not stated in the main text.
>
> The semantic content vector is obtained by feeding each textual payload into a pretrained sentence encoder. We use `sentence-transformers/all-mpnet-base-v2` [1], a Transformer-based encoder trained via contrastive learning, which is strong for semantic similarity and text embedding.
> - Embedding dimensionality: We use 768-dim embeddings, balancing representational quality and computational efficiency.
> - About the (1,128)/(2,128)/(3,128) shapes in logs: These indicate (number of active nodes) × (positional embedding dimension). The node count (1/2/3/…) varies by step, while 128 is fixed. This positional matrix is separate from the 768-dim semantic embedding. We now clarify this explicitly in Appendix A.8 to prevent confusion.
> ---
> > **W3-2.** The ablation visualizations (Figure 4) appear two-dimensional, yet the projection or reduction method (t-SNE, UMAP, PCA, or force-directed) is not described.
>
> Figure 4 is produced by applying PCA to the high-dimensional memory node positions. We reduce node positions from 128 dimensions to 2D, preserving 95% of variance, to visualize structural relations among nodes. This choice and procedure are now stated in the revision.
>
> References
> [1] Song, Kaitao, et al. *MPNet: Masked and Permuted Pre-training for Language Understanding.* NeurIPS 2020.

---

> ### Author Response · Authors · 2025-11-21
> **Response to Reviewer YpTb (Part 3)**
>
> > **W4.** Ablations are shown only visually, without metrics.
> >
> We agree that quantitative ablations strengthen the claim. Due to space limits, the original paper provided only qualitative visualizations. In the revision, we add a quantitative ablation table in Appendix A.7, reporting results on MSC [1] and HotPotQA [2]. We systematically disable each force (node attraction, node repulsion, source-attention pulling, forgetting) and report changes in Mauve, ROUGE-L, F1, and normalized retrieval latency.
>
> |Task|Metric|Full|w/oAttr.|w/oRep.|w/oPull&Fus.|w/oForget|
> |---|---|---|---|---|---|---|
> |MSC|Mauve|23.5|21.2|21.9|21.7|22.5|
> |MSC|ROUGE-L|16.1|14.7|15.1|15.2|15.4|
> |MSC|F1|34|31.2|32|31.5|32.5|
> |MSC|RetrievalLatency↓|1.00×|0.99×|1.02×|1.01×|1.03×|
> |HotPotQA|Mauve|17.8|16.3|16.8|15.8|16.5|
> |HotPotQA|ROUGE-L|35|33.3|33.8|32.7|33.5|
> |HotPotQA|F1|59|55.7|56.5|54.2|56|
> |HotPotQA|RetrievalLatency↓|1.00×|0.97×|1.01×|1.00×|1.02×|
>
>
> Across both tasks, the full four-force MemoryField performs best. Disabling any force reduces performance:  On MSC, dialogue metrics drop by ~1–3 points, with removing node attraction having the largest effect.  On HotPotQA, F1 drops by ~2–5 points, with removing source-attention pulling most damaging to multi-hop reasoning.  Removing forgetting only slightly reduces accuracy, but increases retrieval latency due to memory growth.
>
> > **W5.** Unclear/inconsistent descriptions and captions in the tables.
>
> Thanks for catching these issues. We have corrected them as follows:
> - Table 1: The title mistakenly carried over from an earlier draft; now fixed.
> - Table 2: The “Overall” column is the average over all instances; we clarified this in the caption.
> - Table 3: ROUGE-L follows the main experiment protocol; we expanded the explanation to avoid confusion.
> ---
> > **Q1.** In your Table 4, you include the Reflexion baseline. Please clarify whether you reproduced these results under your own protocol, or adopted them from the source. If taken from the source, provide an exact citation and justify protocol comparability, since in Shinn et al. the metrics are computed over trials and are notably sensitive to agent configuration. In the original Reflexion paper, Figs. 3–4 show stronger curves on AlfWorld/HotPotQA and explicitly test the ReAct + Reflexion configuration. Please explain why the ReAct + Reflexion variant is not reported in your Table 4.
>  >
> The Reflexion results in Table 4 are obtained from our own implementation based on the official codebase. We run all methods within a unified agent framework and adopt task-specific configurations as follows:
> For HotPotQA, we use the standard full-wiki setting and compute EM following the official normalization protocol [2] (lowercasing, punctuation removal, and whitespace normalization). At each agent step, up to K = 5 candidate passages may be retrieved, with up to T = 3 rounds of “reasoning–retrieval” iterations.
> For FEVER, the agent performs up to T = 3 evidence-retrieval steps in a controlled retrieval environment, and SR is measured as label accuracy.
> In AlfWorld, SR indicates whether the agent successfully completes the full action sequence;
> in ScienceWorld, we report the average cumulative reward across all episodes to measure the agent’s multi-step reasoning and tool-use ability.
> Our method is  not designed to compete with ReAct or Reflexion. MemoryField focuses on improving memory organization and management, whereas ReAct and Reflexion emphasize reasoning and interaction strategies (e.g., planning in environments, deciding when to retrieve external knowledge, and updating decisions using feedback).
> Thus, the purpose of Table 4 is not to establish superiority over ReAct or Reflexion as whole-agent systems, but rather to evaluate MemoryField in a controlled setting where the agent backbone and planning framework are fixed, and only the memory mechanism is replaced. ReAct and Reflexion can be integrated with various memory modules—including MemoryField—and are complementary in design.
> We have clarified this point in the revised manuscript to avoid confusion.
>
>
> References
>
> [1] Jang, Jihyoung, Minseong Boo, and Hyounghun Kim. *Conversation Chronicles: Towards Diverse Temporal and Relational Dynamics in Multi-Session Conversations.* EMNLP 2023.
> [2] Yang, Zhilin, et al. *HotpotQA: A Dataset for Diverse, Explainable Multi-hop Question Answering.* EMNLP 2018.

---

### Meta-Review · Area_Chair_xqM4 · 2026-01-02

**Summary:**

The paper proposes "MemoryField," a dynamic spatial memory architecture for LLMs inspired by gravitational fields. The method models memory nodes as particles in a high-dimensional space, governed by forces such as attraction, repulsion, and decay, to manage long-term interaction history. The authors evaluate the method across dialogue, reasoning, and agentic tasks

**Reviewer Concerns:**

- The reviewers generally appreciated the creativity of the "physics-inspired" metaphor (gravitational fields) to model memory dynamics (Reviewers YpTb, pTa2, 2Fqr).

- The initial submission covered a wide range of tasks (dialogue, reasoning, agents).

- The authors were responsive, adding significant new experiments during the rebuttal phase, including comparisons on the LoCoMo benchmark (responding to FVEg) and additional RAG baselines (responding to pTa2).

**Reviewer Scores:**

- A critical concern raised by Reviewer pTa2 (and echoed by others) is the distinction between this system and advanced RAG systems. While the authors argue this is an "episodic memory" system, the mechanics (embedding text chunks + retrieval) are functionally very similar to RAG. In the additional experiments provided during the rebuttal, MemoryField as a standalone memory module did not outperform specialized RAG baselines like HippoRAG. While combining MemoryField with Hybrid RAG showed gains, the standalone value of the complex "gravitational" architecture over state-of-the-art graph/vector retrieval remains unconvincing.

- The proposed architecture introduces significant complexity (four force types, fusion, forgetting, position vs. semantic vectors). Reviewers (2Fqr, YpTb) expressed concern regarding the computational cost and the necessity of such a complex physics simulation for text retrieval. The performance gains, while present, do not decisively justify this added architectural complexity compared to simpler, well-tuned retrieval pipelines or sparse memory structures.

- Reviewer YpTb pointed out significant issues with the initial manuscript regarding the definition of metrics (e.g., Success Rate in HotPotQA), the mapping of datasets to task categories, and inconsistencies in table captions. While the authors clarified these in the rebuttal, the initial lack of rigor raises concerns about the robustness of the reported results.

- Reviewer pTa2 noted that the node granularity (paragraphs/chunks) feels closer to document retrieval than the "Trace Theory" the authors invoke. While the authors added discussion on this, the architectural implementation still feels disjointed from the theoretical claims of modeling human-like episodic traces.

---

### Decision · Program_Chairs · 2026-01-26

Reject